# PRDM16 functions as a co-repressor in the BMP pathway to suppress neural stem cell proliferation

Li He[1], Jiayu Wen[2]*, Qi Dai[1]*

[1]Department of Molecular Bioscience, the Wenner-Gren Institute, Stockholm University, Stockholm, Sweden; [2]Division of Genome Sciences and Cancer, The John Curtin School of Medical Research, The Australian National University and Australian Research Council Centre of Excellence for the Mathematical Analysis of Cellular Systems, Canberra, Australia

## eLife Assessment

This **valuable** work presents how PRDM16 plays a critical role during colloid plexus development, through regulating BMP signaling. **Solid** evidence supports the context-dependent gene regulatory mechanisms both in vivo and in vitro. The work will be of broad interest to researchers working on growth factor signaling mechanisms and vertebrate development.

*For correspondence:
jiayu.wen@anu.edu.au (JW);
qi.dai@su.se (QD)

Competing interest: The authors declare that no competing interests exist.

**Abstract** BMP signaling acts as an instructive cue in various developmental processes such as tissue patterning, stem cell proliferation, and differentiation. However, it is not fully understood how this signaling pathway generates different cell-specific outputs. Here, we have identified PRDM16 as a key co-factor for BMP signaling in the mouse brain. PRDM16 contributes to a repressive role of BMP signaling on neural stem cell (NSC) proliferation. We demonstrate that PRDM16 regulates the genomic distribution of BMP pathway transcription factors, the SMAD4/pSMAD complex, preventing the activation of cell proliferation genes. When *Prdm16* is lost, the SMAD complex relocates to nearby genomic regions, leading to abnormal upregulation of BMP target genes. This function of PRDM16 is also required for the specification of choroid plexus (ChP) epithelial cells. Through a single-cell resolution fluorescent in situ approach, we have observed that genes co-repressed by SMAD and PRDM16, such as *Wnt7b* and several cell cycle regulators, become overexpressed in *Prdm16* mutant ChP. Our findings elucidate a mechanism through which SMAD4 and pSMAD1/5/8 repress gene expression. Moreover, our study suggests a regulatory circuit composed of BMP and Wnt signaling, along with PRDM16, in controlling stem cell behaviors.

## Introduction

Uncontrollable cell proliferation can lead to tumor growth, while premature differentiation can result in tissue degeneration. The balance between stem cell proliferation and differentiation is a crucial aspect in developmental and stem cell biology. BMP (Bone morphogenetic proteins) signaling is a key cell-signaling pathway in regulating stem cell proliferation and maintaining adult stem cell quiescence (*Mira et al., 2010*; *Colak et al., 2008*; *Zhang et al., 2016*). Moreover, BMP signaling is essential in various cell specification processes (*Zhang and Li, 2005*; *Hogan, 1996*; *Massagué, 1998*).

The ability of a single pathway to play a diverse range of roles relies on context-specific transcriptional outputs. BMPs signal through two types of SMAD proteins: the receptor SMADs (R-SMADs) - Smad1, 5, and 8, and the co-SMAD protein SMAD4. Upon ligand binding, heterodimeric receptors

like BMPRI and BMPRII phosphorylate R-SMADs, leading to the assembly and nuclear translocation of the SMAD complex with SMAD4. This complex then regulates gene expression by binding to enhancers of BMP target genes. Apart from transducing BMP signaling, SMAD4 is an essential effector in TGF-β/Activin signaling. In response to ligands such as TGF-β and Activin, SMAD4 associates with two other R-SMADs - phosphorylated SMAD 2 and 3, regulating downstream genes of the TGF-β/Activin signaling pathway. These SMADs recognize and directly bind to two main types of short DNA motifs in target enhancers via their N-terminal MH1 domain (*Martin-Malpartida et al., 2017*). However, since the binding is generally weak, SMAD complexes rely on various co-factors such as transcription factors, co-activators and co-repressors to activate or repress target gene expression (*Hill, 2016*).

In the mammalian brain, BMP and Wnt signaling pattern the brain midline where an essential structure, the ChP, emerges. Neural epithelial cells at the presumptive ChP site lose neural potential and exit the cell cycle. Only a small number of cells at the border of the ChP primordium and cortical hem (CH) persist as slowly dividing ChP progenitors, leading to the expansion of the monolayered ChP epithelial cells (*Liddelow et al., 2010*). BMP signaling is essential for ChP epithelium specification. Conditional depletion of the BMP receptor *BMPr1a* diminishes ChP development (*Hébert et al., 2002*), whereas ectopic BMP transforms neural progenitors into ChP cells (*Watanabe et al., 2012*). Wnt signaling, peaking at CH, is also necessary for ChP epithelial cell specification (*Parichha et al., 2022*). Conditional depletion of ß-Catenin results in defective ChP, and overexpression of ß-Catenin expands CH at the expense of the ChP epithelium. These findings emphasize the importance of tightly controlling Wnt and BMP signaling levels for proper cell specification.

The PR domain-containing (PRDM) family protein PRDM16 determines cell fate specification in various cell types (*Aguilo et al., 2011*; *Chuikov et al., 2010*; *Kajimura et al., 2008*; *Shimada et al., 2017*; *Chui et al., 2020*; *Baizabal et al., 2018*; *Inoue et al., 2017*). Previous studies, including our own, demonstrated that *Prdm16* knockout (KO) mouse brains show severely reduced ChP structures (*He et al., 2021*; *Bjork et al., 2010*; *Strassman et al., 2017*). Interestingly, it was shown that PRDM16 can interact with TGF-ß pathway SMAD proteins in vitro and impact TGF-ß signaling output in craniofacial tissues (*Bjork et al., 2010*; *Warner et al., 2007*). PRDM16 is also a downstream effector of BMP signaling during brown adipocyte specification (*Seale et al., 2007*; *Tseng et al., 2008*). A recent study reported that PRDM16 and its ortholog PRDM3 (also known as Evi1) regulate Wnt signaling during craniofacial development in zebrafish (*Shull et al., 2022*). These findings suggest that PRDM16 may be more broadly involved in regulating BMP/TGF-ß and Wnt signaling. However, the underlying molecular mechanisms remain unclear.

Consistent with its essential roles in normal development, mutations and dysregulation of *Prdm16* are linked with several human diseases, including those identified in patients with 1p36 chromosomal aberrations and cardiomyopathy (*Arndt et al., 2013*). PRDM16 exhibits versatile functions at the molecular level, regulating chromatin accessibility and epigenetic states of its bound enhancers (*Baizabal et al., 2018*; *He et al., 2021*). Depending on associated cofactors, PRDM16 can either repress or activate gene expression. This dual role poses a challenge when considering PRDM16 as a therapeutic target, as an undesired outcome may occur. Thus, a comprehensive understanding of the regulatory roles of this protein in each specific process is crucial for effective disease treatment strategies.

In this study, we have investigated the mechanisms that regulate the transition between stem cell proliferation and differentiation during ChP development. We show that *Prdm16* mutant ChP cells fail to exit the cell cycle, a similar phenotype to when BMP signaling is abolished (*Hébert et al., 2002*). Using primary NSC culture, we dissected the molecular interaction of SMAD4/pSMAD1/5/8 proteins with PRDM16, and found that PRDM16 functions as a co-repressor that holds the SMAD complex at distal enhancers, repressing genes involved in cell proliferation. Finally, we validated that some of the co-regulated genes by BMP signaling and PRDM16 become de-repressed in the *Prdm16* mutant ChP. These findings uncover an essential function of PRDM16 in stem cell regulation and BMP and Wnt signalling.

## Results

### PRDM16 promotes cell cycle exiting of neural epithelial cells at the ChP primordium

To understand how *Prdm16* regulates ChP epithelial specification, we first investigated the expression of this gene in the developing mouse brain. At embryonic day 10.5 (E10.5), BMP signaling specifies the presumptive ChP (*Hébert et al., 2002*). *Prdm16* mRNAs and nuclear localization of the PRDM16 protein become detectable in the presumptive ChP at this stage (*Figure 1A*, *Figure 1—figure supplement 1A*). As previously shown, *Prdm16^cGT^*, a *Prdm16* knockout allele (*Prdm16 KO*), displayed severely reduced ChP structure at E13.5 (*Figure 1B*; *He et al., 2021*; *Bjork et al., 2010*; *Strassman et al., 2017*). Expression of *Ttr*, a classic ChP marker gene, is reduced in both the lateral ventricle (tChPs) (*Figure 1C*) and the fourth ventricle (hChP) (*Figure 1—figure supplement 1B*) in the *Prdm16 KO* mutant, suggesting that the function of PRDM16 is not area-restricted but generally required for ChP development.

To assess the patterning of the CH and ChP regions, we analyzed the expression of *Wnt2b* and *BMP4* using conventional RNA in situ hybridization. *Wnt2b* expression, which marks the CH, appeared comparable between *Prdm16* KO and control brains at E11.5 and E12.5 (*Figure 1—figure supplement 1C*), indicating that CH development is not affected by the loss of *Prdm16*. BMP4, which labels the ChP and CH, also showed normal expression levels and spatial distribution in the mutant brain. These findings suggest that the BMP-dependent patterning of the ChP and CH domains remains intact in the absence of *Prdm16*.

However, the ChP epithelial layer in *Prdm16* mutants remained abnormally thick, resembling the adjacent neural epithelium (*Figure 1C*, *Figure 1—figure supplement 1C*). This observation led us to investigate the underlying cellular changes responsible for the ChP defects in the mutant. To assess cell proliferation, we performed a 2 hr EdU labeling at E12.5. In control embryos, ChP cells were largely EdU negative and formed a monolayer, indicating that most had exited the cell cycle. In contrast, *Prdm16* mutant ChP cells marked by *β-Gal* remained highly proliferative (*Figure 1D*). These results suggest several possibilities: ChP epithelial cells are not properly specified in the mutant, mutant ChP cells are specified but fail to exit the cell cycle to differentiate, or both processes are impaired.

To explore these possibilities, we first examined the cell type composition of *Prdm16* mutant ChP cells at E12.5. We stained brain sections from control and mutant embryos with the neuronal marker Doublecortin (Dcx). In controls, Dcx-positive cells were absent from the ChP, indicating that neural epithelial cells had successfully transitioned into ChP epithelial cells and lost their neural potential (*Figure 1E*). In contrast, the *Prdm16* mutant ChP exhibited numerous Dcx-positive cells along the basal side of the epithelium, suggesting ongoing neurogenesis.

We next examined the expression of the NSC marker SOX2. In control brains, SOX2 is highly expressed in NSCs adjacent to the ChP epithelium but is significantly downregulated within the presumptive ChP region (*Figure 1—figure supplement 1D*). This downregulation was absent in the *Prdm16* mutant. Together, these results indicate that at E12.5, *Prdm16* depletion disrupts the normal transition of neural progenitors into ChP epithelial cells, maintaining cells in a proliferative, undifferentiated state.

We then asked whether PRDM16 not only regulates ChP epithelial specification but also directly restricts cell proliferation. To test this, we examined *Prdm16* mutant ChP cells at a later stage. We performed double S-phase labeling by injecting EdU at E12.5 and BrdU at E13.5, followed by sample collection at E14.5. By this stage, the wild-type ChP had developed into an elongated, monolayered epithelium in which most cells were negative for both EdU and BrdU, indicating cell cycle exit. In contrast, the *Prdm16* mutant ChP remained a disorganized cluster of cells, many of which were EdU-positive, BrdU-positive, or both (*Figure 1F–G*). This finding suggests that although the mutant ChP can separate from the neuroepithelium at later stages, it fails to exit the cell cycle and differentiate, supporting a direct role of PRDM16 in restricting proliferation.

Given that BMP signaling is a key regulator of ChP formation and that the BMP pathway mutants, such as *Foxg1^cre^::Bmpr1a^fl/fl^* also exhibit ectopic cell proliferation in the developing ChP (*Hébert et al., 2002*), we next investigated whether PRDM16 interacts with the BMP pathway to control cell proliferation.

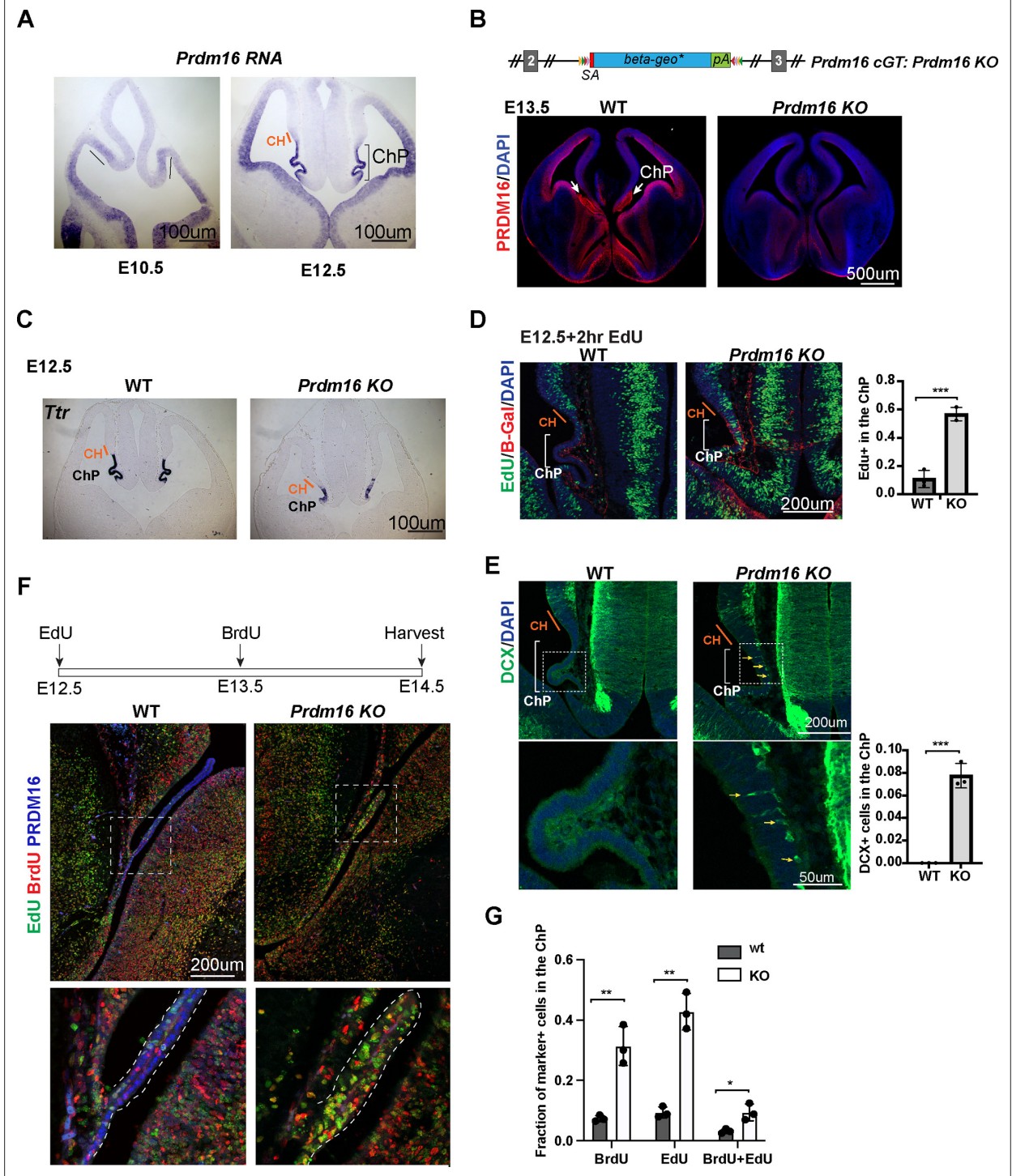

**Figure 1.** *Prdm16* mutant choroid plexus (ChP) cells fail to exit the cell cycle and retain neural progenitor identity. (**A**) RNA in situ hybridization using a *Prdm16* probe shows *Prdm16* expression in E10.5 and E12.5 brains. The line and bracket mark the developing CH and ChP. (**B**) Top: Schematic of the *Prdm16* GT allele, as reported in *Strassman et al., 2017*. Bottom: immunostaining of PRDM16 in E13.5 control and null mutant brains. Red: PRDM16; blue: DAPI. White arrows indicate the ChP in the control brain. (**C**) RNA in situ hybridization for *Ttr* on E12.5 control and *Prdm16* null mutant forebrains. Cortical hem (CH) regions are outlined based on morphology. (**D**) Two-hour EdU labeling in E12.5 control and *Prdm16* homozygous brains. Sections are co-stained with anti-*β*-Gal antibody to identify mutant ChP cells in the KO brain. Note: the *β-Geo* trap is absent in wild-type animals. Brackets mark ChP regions. Red: β-Gal; blue: DAPI; green: EdU. (**E**) Immunostaining for Doublecortin (DCX) on E12.5 control and *Prdm16* homozygous brain slices. Bottom panels show magnified views of the ChP (dashed rectangles). Yellow arrows indicate some of the DCX-positive cells. Right panels show quantification of EdU+ (**D**) and DCX+ cells (**E**). Biological replicates: N=3. Error bars represent standard deviation (SD). Statistical significance is calculated using

*Figure 1 continued on next page*

*Figure 1 continued*

unpaired t-test. ***p<0.001. (**F**) Schematic of the double S-phase labeling experiment (top). Brain sections from E14.5 control and prdm16 KO embryos were immunostained with PRDM16 and BrdU antibodies and processed for EdU detection. Bottom panels show magnified views of the ChP region. (**G**) Quantification of EdU+ and BrdU+ cells from the double labeling experiment. Biological replicates: N=3. Error bars represent standard deviation (SD). Statistical significance is calculated using unpaired t-test. **p<0.01. *p<0.05.

The online version of this article includes the following source data and figure supplement(s) for figure 1:

**Source data 1.** Counting of EdU+ and DAPI stained cells, related to *Figure 1D*.

**Source data 2.** Counting of Dcx+ and DAPI stained cells, related to *Figure 1E*.

**Source data 3.** Counting of EdU+, BrdU+, double positive and DAPI stained cells, related to *Figure 1G*.

**Figure supplement 1.** *Prdm16* depletion affects choroid plexus (ChP) epithelial specification without altering dorsal midline patterning.

**Figure supplement 1—source data 1.** Measurement of SOX2 signal intensity, related to *Figure 1—figure supplement 1D*.

## PRDM16 and BMP signaling collaborate to induce NSC quiescence in vitro

To investigate the molecular interplay between PRDM16 and BMP signaling in regulating cell proliferation, we turned to primary NSCs as an in vitro model. BMP signaling is known to maintain quiescence in adult NSC in vivo and can induce proliferative NSCs into quiescence in vitro (*Mira et al., 2010*). Furthermore, previous studies have shown that embryonic cortical NSCs are responsive to BMP4 treatment (*Hu et al., 2008*), making them a suitable system for probing downstream BMP signaling events. Based on this, we established a cell culture assay to evaluate the effects of BMP4 and PRDM16 on NSC proliferation and quiescence (*Figure 2A*).

Unexpectedly, we found that PRDM16 protein levels were undetectable in cultured NSCs despite high levels of *Prdm16* mRNAs (*Figure 2—figure supplement 1A–B*). Given that the PRDM16 protein is normally restricted to the nucleus of NSCs in embryonic brain tissues (*Figure 1B*, *Figure 1—figure supplement 1A*; *He et al., 2021*), this observation suggests that in vivo mechanisms may regulate PRDM16 protein stability and nuclear localization. Post-translational modification of PRDM16 has been identified as a key regulatory mechanism in brown adipocytes (*Wang et al., 2022*), and we speculate that a similar mechanism may operate in NSCs.

To bypass this limitation and examine PRDM16's molecular function in regulating NSC proliferation and gene regulation, we generated a lentiviral construct expressing *3xNSL_Flag_Prdm16* under a constitutive promoter. Infection of wild-type primary NSCs with this construct yielded a cell line with robust *Prdm16* mRNA expression and detectable nuclear PRDM16 protein (*Figure 2—figure supplement 1A–B*), which we referred to as *Prdm16_overexpressing (Prdm16_OE)*. For comparison, we attempted to establish three additional lines: wild-type NSCs infected with the empty vector (*wt_CDH*), *Prdm16_KO* NSCs infected with the empty vector (KO_CDH), and *Prdm16_KO* NSCs infected with *3xNSL_Flag_Prdm16*. However, we were unable to generate the last line despite repeated attempts. This failure was likely due to low viral production (the *Prdm16* coding sequence exceeds 3 kb) and the increased sensitivity of KO NSCs to viral infection. Nevertheless, we proceeded with comparative analyses of *Prdm16_OE* cells against *wt_CDH*, *KO_CDH,* and uninfected *KO* NSCs.

To assess cell proliferation rates, we labeled NSCs with EdU and mKi67. BMP4 treatment of *Prdm16_OE* NSCs led to a marked reduction in the number of mKi67- and EdU-positive populations (*Figure 2B–D*). Following BMP4 washout, *Prdm16_OE* NSCs restored EdU and mKi67 labeling, indicating re-entry into the cell cycle. This reversible reduction confirms that BMP4 induces a quiescent, non-proliferative state in NSCs. In contrast, a larger proportion of *Prdm16_KO* cells failed to exit the cell cycle in response to BMP4, as shown by a less pronounced decrease in mKi67- and EdU-positive populations (*Figure 2B–D*). Notably, *Prdm16_KO* cells exhibited similar properties regardless of whether they were infected with the control viral vector (*Figure 2—figure supplement 1D–F*), and we, therefore, used both lines interchangeably in our analyses.

Previous studies have shown that BMP4 can program ChP cell fate and activate ChP-specific genes in neural progenitors cultured in vitro (*Watanabe et al., 2012*). To test whether PRDM16 cooperates with BMP signaling to induce ChP gene expression, we measured *Ttr* mRNA levels using reverse transcription followed by quantitative PCR (RT-qPCR). In *Prdm16_OE* NSCs, *Ttr* expression was robustly induced following BMP4 treatment. In contrast, *Prdm16_KO* NSCs and NSCs lacking the *Prdm16_OE* construct failed to upregulate *Ttr* in response to BMP4 (*Figure 2E*, *Figure 2—figure supplement 1G*).

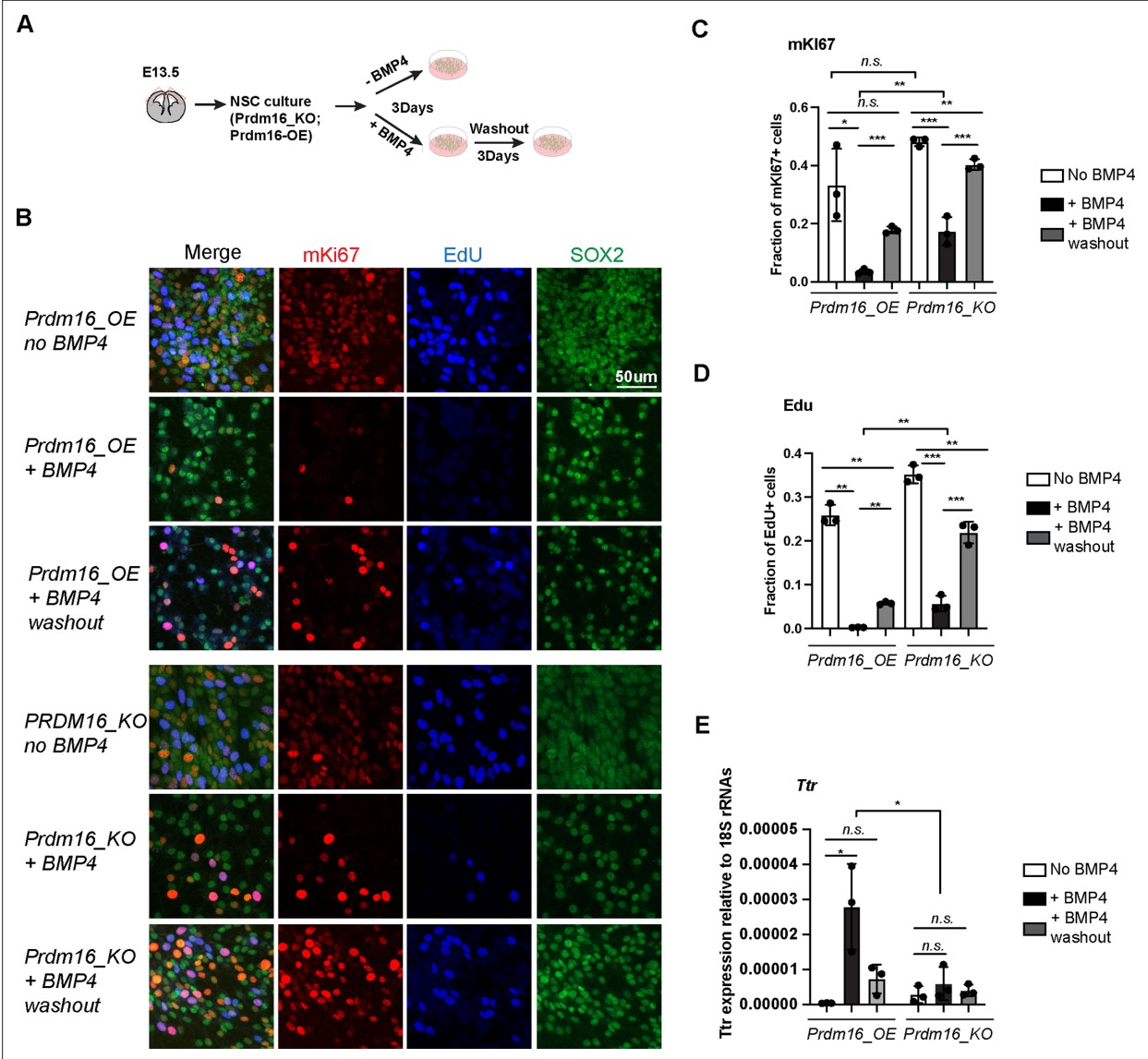

**Figure 2.** *Prdm16* is required for BMP4-induced neural stem cell (NSC) quiescence. (**A**) Schematic of the NSC culture assay. (**B**) Immunostaining of mKI67 in red and SOX2 in green and EdU labeling in blue on indicated NSC genotypes and treatment. (**C–D**) Quantification of the fraction of mKI67+ (**C**) and EdU+ (**D**) cells among the total cell number marked by SOX2. (**E**) RT-qPCR measurement of *Ttr* levels relative to 18 S RNAs. Biological replicates: N=3. Error bars represent standard deviation (SD). Statistical significance is calculated using unpaired t-test. ***p<0.001; **p<001; *p<0.05; n.s., non-significant.

The online version of this article includes the following source data and figure supplement(s) for figure 2:

**Source data 1.** Counting of mKI67 + and DAPI-stained cells, related to *Figure 2C*.

**Source data 2.** Counting of EdU+ and DAPI-stained cells, related to *Figure 2D*.

**Source data 3.** RTqPCR measurement of Ttr and 18srRNA expression, related to *Figure 2E*.

**Figure supplement 1.** Neural stem cell (NSC) culture assay reveals PRDM16 function in NSCs.

**Figure supplement 1—source data 1.** RTqPCR measurement of Prdm16 and 18srRNA expression, related to *Figure 2—figure supplement 1B*.

**Figure supplement 1—source data 2.** Counting of mKI67+ and DAPI-stained cells, related to *Figure 2—figure supplement 1D*.

**Figure supplement 1—source data 3.** Counting of EdU+ and DAPI-stained cells, related to *Figure 2—figure supplement 1E*.

**Figure supplement 1—source data 4.** RTqPCR measurement of Ttr and 18srRNA expression, related to *Figure 2—figure supplement 1G*.

Together, these results indicate that both BMP signaling and PRDM16 are required not only to restrict NSC proliferation but also to induce ChP gene expression. We next investigated the molecular mechanisms underlying this cooperation.

## BMP signaling and PRDM16 cooperatively repress proliferation genes

To understand how BMP signaling and PRDM16 suppress cell proliferation, we aimed to determine the transcriptional targets of SMAD4 and pSMAD1/5/8 in cells with and without BMP4. We first applied Cleavage Under Targets and Tagmentation (CUT&TAG) experiments using a PRDM16 antibody and the available SMAD antibodies to profile their genomic binding sites. The PRDM16 antibody worked efficiently, but none of the SMAD antibodies produced libraries with sufficient sequencing reads. Subsequently, we employed chromatin immunoprecipitation followed by deep sequencing (ChIP-seq) for the SMAD proteins. Given that SMAD4 forms a complex with SMAD2/3 only in response to TGF-β/activin-type ligands, we included an antibody targeting SMAD3 as a control for non-BMP4-induced SMAD4 binding. Thus, our experiment settings enabled us to profile genomic binding sites for PRDM16, SMAD4, and two types of R_SMADs under conditions with endogenous BMPs and TGF-β/Activin, as well as ectopic BMP4.

In *Prdm16_OE* cells without BMP4, we identified several hundred to a few thousand ChIP-seq peaks for all three classes of SMAD proteins after normalizing to the input reads (FDR <10%), indicating endogenous levels of BMP and TGF-β signaling in these cells (*Figure 3A*, *Figure 3—figure supplement 1A–C*). Following BMP4 addition, pSMAD1/5/8 exhibited approximately a 750-fold increase in peak number, while the number of SMAD3 peaks increased by less than threefold (*Figure 3A*, *Figure 3—figure supplement 1A*). Furthermore, in cells treated with BMP4, the pSMAD1/5/8 peaks largely overlap with the SMAD4 peaks, but much less with the SMAD3 peaks (*Figure 3—figure supplement 1B*). For example, pSMAD1/5/8 binds to two enhancers in the intronic region of *Prdm16*, with the binding intensity dramatically increased in response to BMP4 (*Figure 3—figure supplement 1D*). This suggests that *Prdm16* itself is a transcriptional target of BMP signaling. Indeed, *Prdm16* mRNA levels were elevated by BMP4 in wild-type but not *Prdm16_KO* cells (*Figure 2—figure supplement 1B*). By contrast, SMAD3 showed little ChIP-seq signal in the *Prdm16* gene locus even in the presence of BMP4. These results confirm that the response of pSMAD1/5/8 is specific to BMP4, while SMAD3 is generally unresponsive to BMP4. The gained Smad3 sites likely result from an indirect effect of BMP signaling.

To evaluate the regulatory activity of SMAD4/pSMAD1/5/8 in NSCs, we determined the state of transcription activation using H3K4me3 CUT&TAG signals at annotated gene transcription start sites (TSS) in BMP4-treated and untreated *Prdm16_OE* cells. As cell identity genes are known to be marked by extended breadth of H3K4me3 (*Benayoun et al., 2014*), we reasoned that compared to RNA-seq, which measures all gene products, changes in the H3K4me3 signal at the promoter might bias toward cell identity genes between proliferation and quiescence states. Thus, we determined all BMP4-repressed and activated genes by measuring changes in the H3K4me3 CUT&TAG read coverage at annotated TSS in BMP4-treated *Prdm16_OE* cells compared to untreated, identifying 282 up-regulated and 429 down-regulated genes (*Figure 3B*) (p<0.01). To pinpoint genes directly regulated by SMAD4 and pSMAD1/5/8, we intersected the dysregulated genes with those whose regulatory regions contain overlapping SMAD4 and pSMAD1/5/8 ChIP-seq peaks. This analysis led to the identification of 145 up-regulated and 184 down-regulated SMAD4 and pSMAD1/5/8 target genes.

To elucidate the function of BMP-regulated targets, we performed gene ontology (GO) analyses for the genes that changed expression in response to BMP4 in *Prdm16_OE* cells and were also bound by SMAD4 and pSMAD1/5/8. Remarkably, the downregulated genes (BMP4-repressed genes), but not the up-regulated genes (BMP4-activated genes), showed significantly enriched functional categories, with nearly all GO terms related to cell proliferation/cell cycle (*Figure 3C*). A recent study reported that BMP2-induced genes are enriched for neuronal and astrocyte differentiation (*Katada et al., 2021*), while our analysis did not identify these categories. One possibility for the discrepancy is that we overexpress *Prdm16* in cultured NSCs, which may reinforce BMP signaling activities in cell proliferation. Other possibilities could be the use of different BMP ligands (BMP4 versus BMP2), differences in the origin of NSC culture (ours was from E13 instead of E11 and E14), or different profiling methods (H3K4me3 versus RNA-seq).

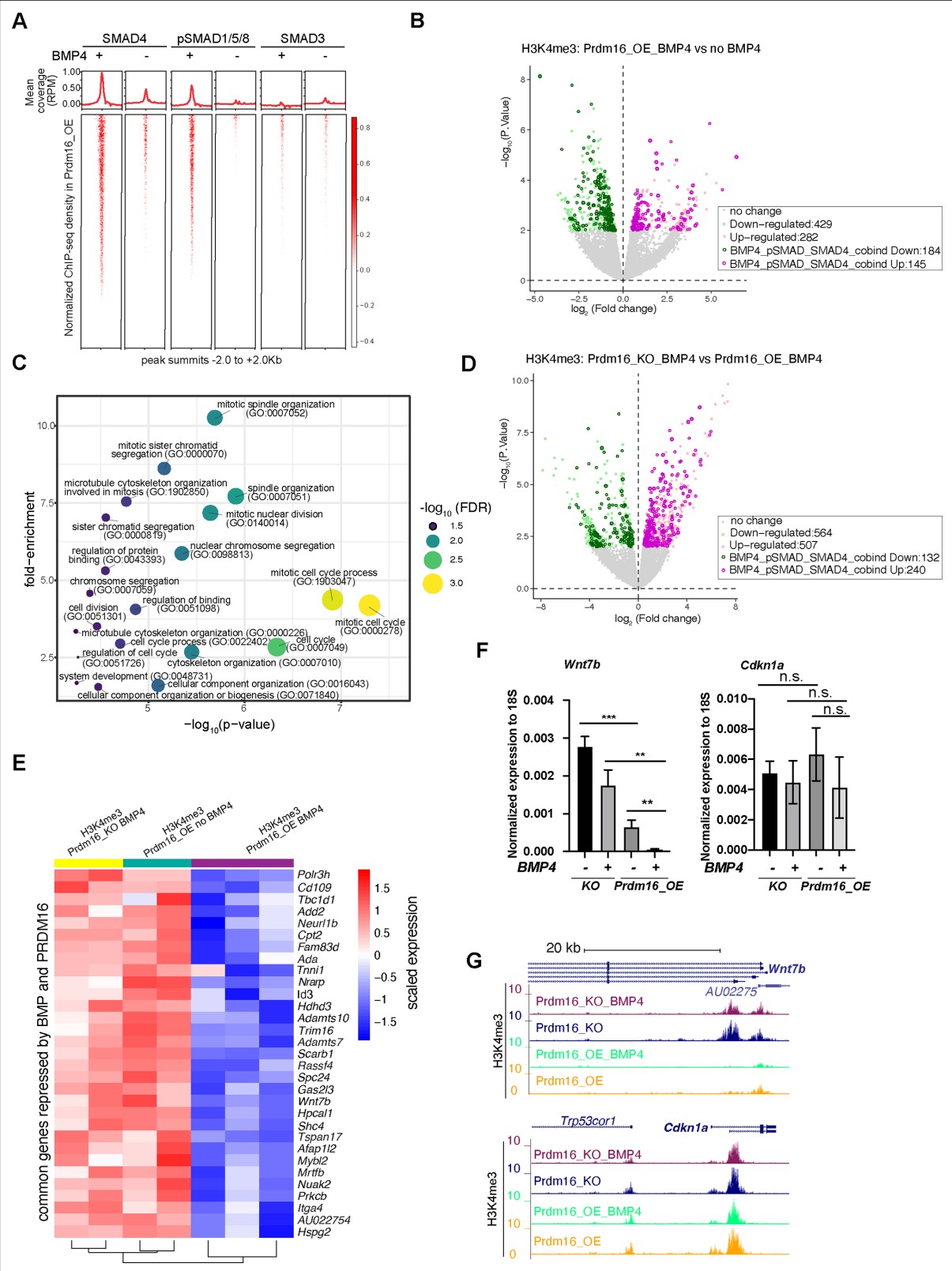

**Figure 3.** BMP signaling represses cell proliferation genes to induce cell quiescence. (**A**) Heatmaps of ChIP-seq coverage (normalized by library depths and input) centered at peak summits for SMAD4, pSMAD1/5/8 and SMAD3 in *Prdm16_OE* cells with or without BMP4. (**B**) Volcano plot displaying differential H3K4me3 signal at TSS in *Prdm16_OE* cells with BMP4 versus untreated controls. Genes with significant changes (p<0.01) are colored. (**C**) Bubble plot of GO terms significantly enriched among SMAD-repressed genes (highlighted in dark green in panel B). (**D**) Volcano plot of differential

*Figure 3 continued on next page*

*Figure 3 continued*

H3K4me3 signal at TSS comparing *Prdm16_KO* and *Prdm16_OE* cells, both treated with BMP4. Significantly changed genes (p<0.01) are colored. (**E**) Heatmap of normalized H3K4me3 read counts at the transcription start sites (TSS) of 31 genes co-repressed by PRDM16 and SMADs (overlapping genes from dark green in B and dark pink in D). (**F**) RT-qPCR quantification of *Wnt7b* and *Cdkn1a* mRNA levels across indicated genotypes and treatment conditions. Data represent three biological replicates, each with two technical replicates. Statistical significance was calculated using unpaired t-test (***p<0.001; **p<0.01; n.s., non-significant). (**G**) Genome browser tracks of H3K4me3 CUT&TAG profiles at the *Wnt7b* and *Cdkn1a* loci.

The online version of this article includes the following source data and figure supplement(s) for figure 3:

**Source data 1.** RTqPCR measurement of *Wnt7b*, *Cdkn1a,* and 18srRNA expression, related to *Figure 3F*.

**Figure supplement 1.** Overlapping analysis of SMAD4, pSMAD1/5/8, and SMAD3 CUT&TAG peaks.

**Figure supplement 1—source data 1.** RT-qPCR measurement of *Mybl2* and 18srRNA expression, related to *Figure 3—figure supplement 1F*.

**Figure supplement 1—source data 2.** RT-qPCR measurement of Id3 and 18srRNA expression, related to *Figure 3—figure supplement 1G*.

**Figure supplement 1—source data 3.** RT-qPCR measurement of Spc25 and 18srRNA expression, related to *Figure 3—figure supplement 1I*.

**Figure supplement 1—source data 4.** RT-qPCR measurement of Ndc80 and 18srRNA expression, related to *Figure 3—figure supplement 1J*.

**Figure supplement 1—source data 5.** RT-qPCR measurement of Nuf2 and 18srRNA expression, related to *Figure 3—figure supplement 1K*.

To identify PRDM16-repressed and activated genes under active BMP signaling conditions, we compared changes in H3K4me3 coverage at TSSs between BMP4-treated *Prdm16_KO* cells and BMP4-treated *Prdm16_OE* cells. We found approximately twice as many up-regulated genes (240) as down-regulated genes (132) (*Figure 3D*), suggesting that the cooperative activity of PRDM16 and BMP signaling mainly represses gene expression. We further overlapped the 184 genes repressed by BMP4 and the 240 genes repressed by PRDM16, identifying 31 common ones. These 31 genes displayed low H3K4me3 coverage in *Prdm16_OE* cells with BMP4 but higher H3K4me3 signal in both *Prdm16_KO* cells treated with BMP4 and untreated *Prdm16_OE* cells (*Figure 3E*).

Next, we attempted to validate whether changes in TSS H3K4me3 intensity correspond to changes in mRNA levels, by conducting RT-qPCR for selected genes whose function is associated with cell proliferation from the gene ontology analysis: *Wnt7b*, *Mybl2*, *Id3*, *Spc24* and the three *Spc24*-related genes (*Spc25*, *Ndc80* and *Nuf2*) (*Figure 3F–G*, *Figure 3—figure supplement 1E–K*). SPC24 forms the NDC80 kinetochore complex along with three other proteins: SPC25, NDC80, and NUF2 (*Tooley and Stukenberg, 2011*) (Illustrated in *Figure 3—figure supplement 1H*), and their function is essential for chromosome segregation and spindle checkpoint activity. Notably, *Spc25*, *Ndc80,* and *Nuf2* appeared repressed by PRDM16 and BMP4 based on changes in H3K4me3 at the TSS (*Figure 3—figure supplement 1E*), despite not being identified as the top 31 candidates. mRNA levels for most of the tested genes followed a similar pattern to the H3K4me3 intensity: BMP4-treated *Prdm16_OE* cells showed the lowest expression, while either the loss of *Prdm16* or absence of BMP4 led to upregulation. *Spc24* gene products showed minimal amplification in qPCR, likely due to poor primer sequence quality or unstable mRNAs. Another exception is *Id3* whose expression increased upon BMP4 treatment or *Prdm16* depletion, indicating PRDM16 repressing *Id3* but BMP4 inducing *Id3*. However, the activation of *Id3* by BMP4 is significantly stronger in *Prdm16_KO* cells than that in *Prdm16_OE* cells, suggesting that BMP signaling normally activates *Id3*, but PRDM16 suppresses such activation. In contrast, *Cdkn1a*, a target gene of TGF-β pathway encoding the cell cycle inhibitor P21, exhibited consistent H3K4me3 coverage and mRNA levels across BMP4-treated and non-treated cells, as well as between *Prdm16_OE* and *KO* cells (*Figure 3F–G*), confirming that neither PRDM16 nor BMP signaling influences *Cdkn1a* expression in NSCs.

## PRDM16 assists genomic binding of SMAD4 and pSMAD1/5/8

To understand how PRDM16 interacts with BMP signaling, we integrated PRDM16 CUT&TAG data with SMAD ChIP-seq data, focusing on assessing PRDM16's influence on the genomic binding of SMAD proteins.

The SEACR peak caller software (*Meers et al., 2019*) identified 3337 and 7639 PRDM16 CUT&TAG common peaks from four replicates of BMP4-treated and untreated *Prdm16_OE* samples (FDR <10%, see Methods), respectively (*Figure 4A*). Upon BMP4 treatment, there were 5936 lost and 1634 gained PRDM16 CUT&TAG peaks, suggesting that PRDM16 regulates a distinct subset of genes in proliferating versus quiescent NSCs. Samples from *Prdm16_KO* cells lacked these sites, confirming the specificity of the PRDM16 CUT&TAG signal (*Figure 4B*).

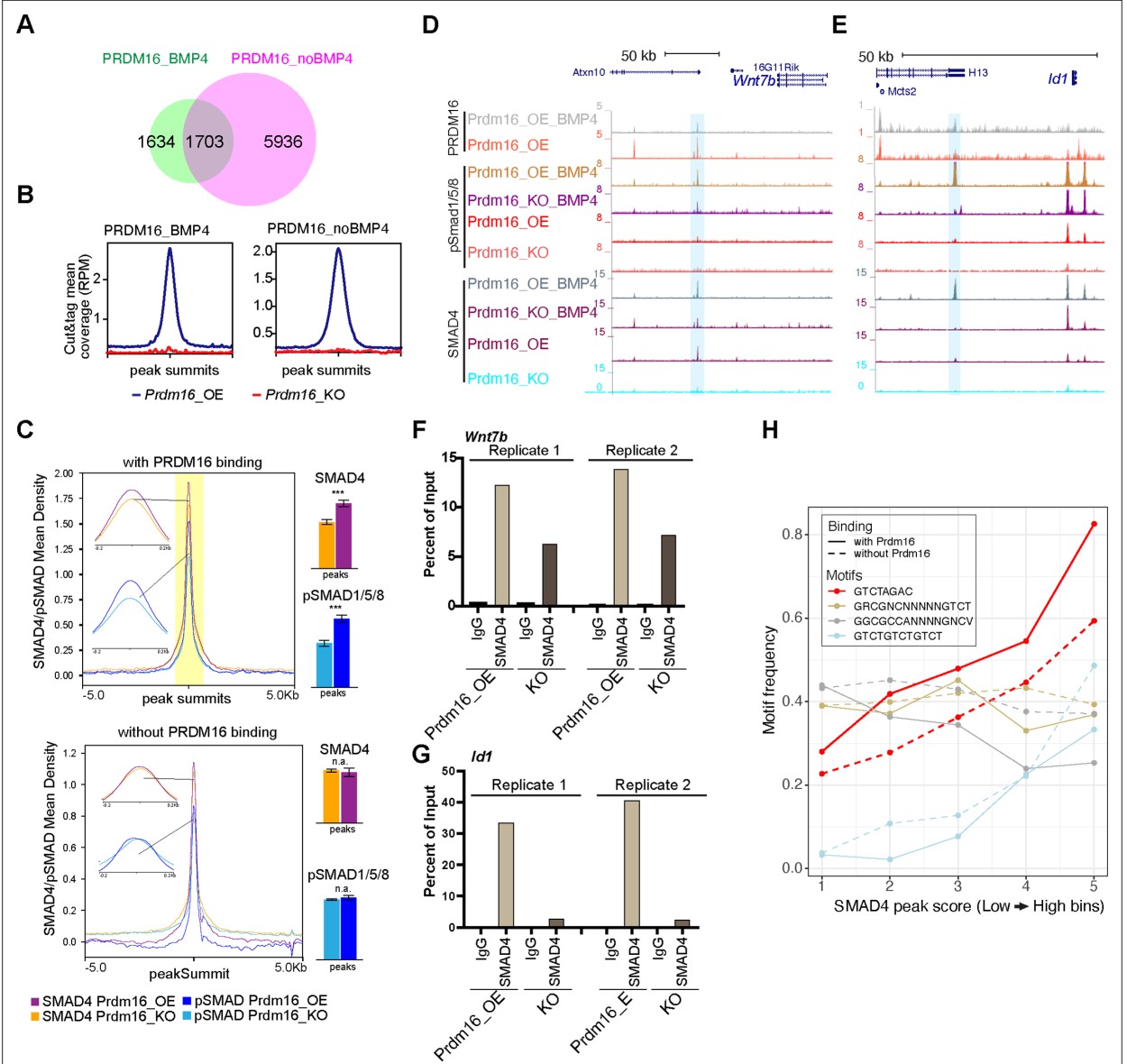

**Figure 4.** PRDM16 anchors SMAD proteins at specific genomic regions to mediate gene repression. (**A**) Venn diagram showing overlap of PRDM16 CUT&TAG peak in BMP4-treated and untreated *Prdm16_OE* cells. (**B**) Average PRDM16 CUT&Tag signal (normalized to library size) at peak summits in BMP4-treated, untreated Prdm16_OE, and Prdm16_KO cells (4 replicates combined). (**C**) Metaplots of SMAD4 and pSMAD1/5/8 ChIP-seq coverage centered on peak summits with (top) and without (bottom) PRDM16 co-binding in BMP4-treated *Prdm16_OE* and *Prdm16_KO* cells. Bar plots (right) show average coverage; significant reduction in the KO cells is seen only at PRDM16 co-bound sites (***p<0.001; **p<0.01; *p<0.05; n.a. p>0.05). (**D–E**) Genome browser view of *Wnt7b* and *Id1* loci showing PRDM16 CUT&TAG and SMAD ChIP-seq tracks. Co-bound regions are highlighted. (**F–G**) ChIP-qPCR validation of SMAD4 binding at highlighted regions in D-E from *Prdm16_OE* and *Prdm16_KO* cells (2 replicates). (**H**) Frequency of four SMAD motif types across SMAD ChIP-seq peaks binned by binding strength. The palindromic motif GTCTAGAC is most enriched at PRDM16 co-bound sites (solid red) and notably reduced in non-co-bound peaks (dashed red).

The online version of this article includes the following source data and figure supplement(s) for figure 4:

**Source data 1.** ChIP-qPCR measurement of SMAD4 and PRDM16 co-bound site in *Wnt7b*, related to *Figure 4F*.

**Source data 2.** ChIP-qPCR measurement of SMAD4 and PRDM16 co-bound site in *Id1*, related to *Figure 4G*.

**Figure supplement 1.** PRDM16 assists SMAD4/pSMAD1/5/8 genomic binding to their co-bound sites.

**Figure supplement 1—source data 1.** ChIP-qPCR measurement of SMAD4 and PRDM16 co-bound sites in the sequential ChIP experiment, related to *Figure 4—figure supplement 1D*.

From our overlapping analyses (see Methods), we found that over 50% of SMAD4 and pSMAD1/5/8 binding peaks were consistent in *Prdm16_OE* and *Prdm16_KO* cells (*Figure 4—figure supplement 1A–B*), indicating that deletion of *Prdm16* does not affect general genomic binding ability of these proteins. Using Homer's mergePeaks function with PRDM16 CUT&TAG and SMAD ChIP-seq data, we identified co-bound sites by PRDM16 and SMAD proteins in BMP4-treated *Prdm16_OE* cells. PRDM16 CUT&TAG peaks mainly overlap with SMAD4 and pSMAD1/5/8 peaks, but not much with SMAD3 peaks (*Figure 4—figure supplement 1A–C*). This result suggests that PRDM16 mainly collaborates with the SMAD4/pSMAD1/5/8 complex but not the SMAD3/SMAD4 complex in cells with high levels of BMP4.

Further examination of SMAD4 and pSMAD1/5/8 binding revealed significantly lower enrichment of SMAD4 and pSMAD1/5/8 at PRDM16 co-bound sites in *Prdm16_KO* cells compared to *Prdm16_OE* cells (*Figure 4C*) (p=2.5E-6, and p=4.7E-3, respectively, two-tailed *t*-test). As a control, the SMAD4 and pSMAD1/5/8 sites without PRDM16 co-binding did not show such a change (p>0.05). This result suggests that SMAD binding to the PRDM16 co-binding sites depends on PRDM16.

To validate the co-binding of SMAD4 and PRDM16, we selected the candidate loci, *Wnt7b* and *Id1* (*Figure 4D–E*) and applied sequential ChIP-qPCR. This experiment confirmed simultaneous binding of SMAD4 and PRDM16 to the same DNA molecules at these loci (*Figure 4—figure supplement 1D*), as SMAD4 was more enriched in the chromatin pulled by the PRDM16 antibody compared to the IgG control. Additionally, ChIP-qPCR confirmed reduced SMAD binding at the SMAD/PRDM16 co-bound site in *Prdm16_KO* cells compared to *Prdm16_OE* cells (*Figure 4F–G*). Thus, PRDM16 enhances genomic binding of the SMAD proteins to specific genome regions.

## PRDM16 facilitates SMAD4 binding to regions enriched for SMAD palindromic motifs

We then wondered whether there is a sequence feature that distinguishes the regions co-bound by SMADs and PRDM16 from those only bound by SMADs. To this end, we checked SMAD-bound regions with PRDM16 binding and those without for two types of SMAD4 binding motifs (*Hill, 2016*). Together with pSMAD1/5/8 or pSMAD3, SMAD4 may bind to a palindromic motif, GTCTAGAC or direct repeats of GTCT like GTCTGTCTGTCT (*Zawel et al., 1998*; *Dennler et al., 1998*; *Wong et al., 1999*); together with pSMAD1/5/8, SMAD4 associates with GC-rich SBEs (SMAD-binding elements) including GGCGCC-AN4-GNCV and GRCGNCNNNNNGTCT (*Pyrowolakis et al., 2004*; *Gao et al., 2005*; *Weiss et al., 2010*). We calculated the occurrence frequency of each of these motifs in binned SMAD4 ChIP-seq peaks from *Prdm16_OE* cells treated with BMP4 (bin 1–5 with low to high peak scores, *Figure 4H*). Interestingly, the palindromic motif is the most enriched one. Its occurrence frequency increases with peak scores but becomes lower in the SMAD4 peaks absent for PRDM16 co-binding. The frequency of the GTCT-triplet motif also increases with higher SMAD4 peak scores, while there is no reduction in regions without PRDM16 binding. The two GC-rich motifs show a distinct trend: they are not as highly enriched, their frequencies do not increase with higher SMAD4 peak scores, and the occurrence frequency in PRDM16 co-bound regions is either comparable to or even lower than the regions without PRDM16 binding. Together, this result suggests that PRDM16 may separate the SMAD4/pSMAD1/5/8 proteins into two types of genomic regions, one enriched for the palindromic motif where PRDM16 is present and the other with the GC-rich motifs.

## SMAD4 and pSMAD1/5/8 switch genomic locations in the absence of PRDM16

Our careful inspection of the *Wnt7b* and *Id1* loci surprisingly revealed that there are multiple ectopic SMAD4 and pSMAD1/5/8 peaks in the *Prdm16* mutant sample (arrow-indicated peaks in *Figure 5A–B*). We then globally assessed the extent of ectopic SMAD binding surrounding the SMAD/PRDM16 co-bound sites by quantifying differential SMAD binding intensity between *Prdm16_KO* and *Prdm16_OE* cells (*Figure 5C*, FDR <10%). In agreement with the metaplots (*Figure 4C*), all of the SMAD/PRDM16 co-bound sites (blue dots) showed reduced ChIP-seq read coverage in *Prdm16_KO* cells. By contrast, the flanking regions within the 200 kb range of a co-bound site (red dots) displayed a trend of increase in SMAD binding in *Prdm16_KO* cells. This result agrees with the aforementioned motif analysis that PRDM16 helps to localize the SMAD complex at specific genomic regions, and it

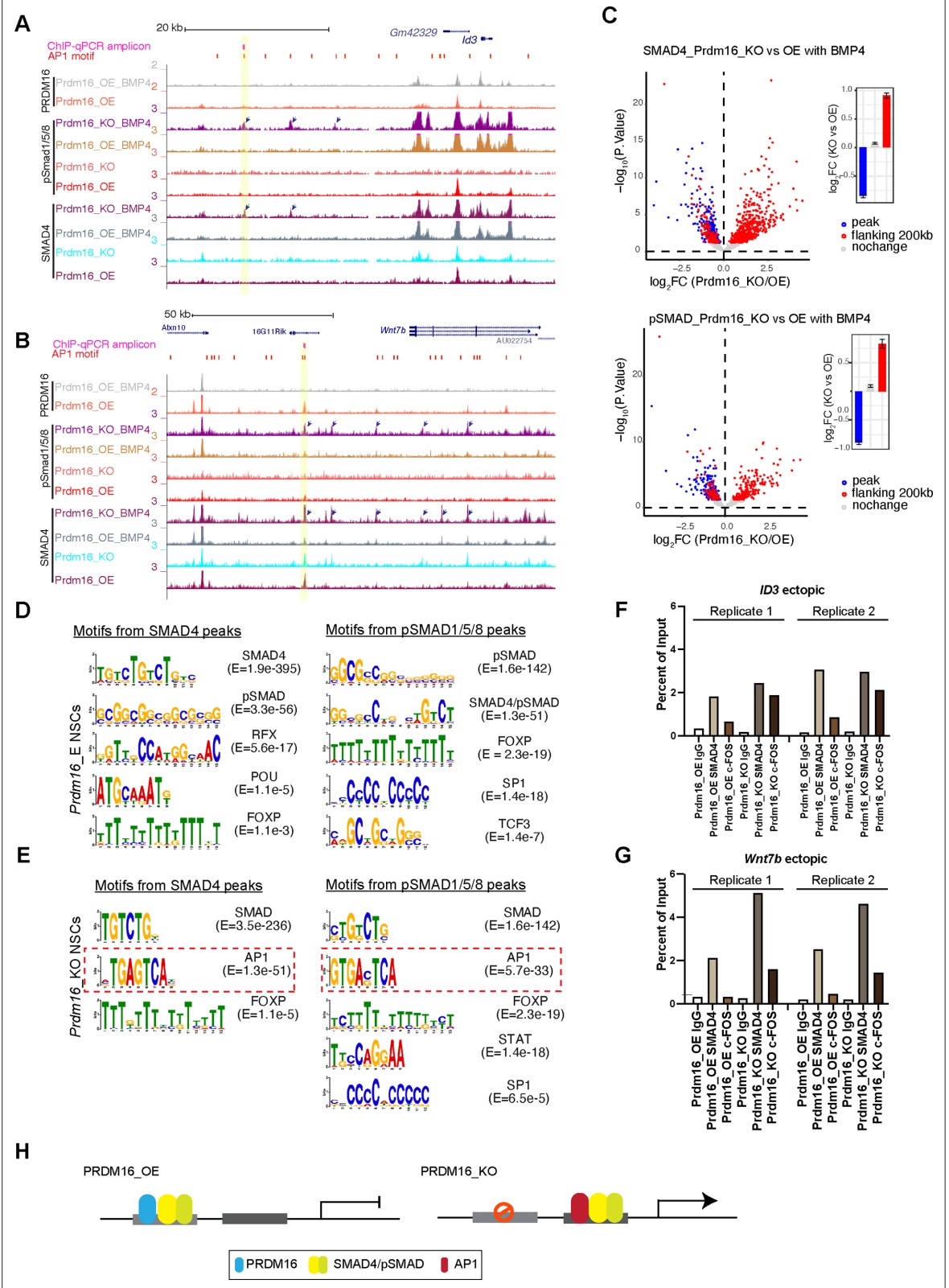

**Figure 5.** The SMAD complex shifts genome binding in the absence of Prdm16. (**A–B**) Genome browser views of the *Id3* (top) and *Wnt7b* (bottom) loci showing ectopic SMAD peaks (arrows) in *Prdm16 KO* cells. (**C**) Volcano plots showing differential ChIP-seq signal for SMAD4 and pSMAD1/5/8 in BMP4-treated *Prdm16_KO* cells versus BMP4-treated *Prdm16_OE* cells. Blue dots represent sites co-bound by PRDM16; red dots indicate sites not bound by PRDM16 but located within 200 kb of a PRDM16-cobound region. Mean log2 fold-change values are plotted to the right. (**D–E**) De novo motif discovery

*Figure 5 continued on next page*

*Figure 5 continued*

from SMAD4 and pSMAD1/5/8 ChIP-seq peaks in *Prdm16_OE* cells (**D**) and *Prdm16_KO* cells (**E**). (**F–G**) ChIP-qPCR validation of increased SMAD4 and c-FOS occupancy at the indicated ectopic binding sites in *Prdm16_KO* versus *Prdm16_OE* cells. Locations of PCR amplicons are shown in A and B. (**H**) Model illustrating enhancer switching: in the absence of PRDM16, SMAD complexes alter their co-factors and function as transcriptional activators.

The online version of this article includes the following source data for figure 5:

**Source data 1.** ChIP-qPCR measurement of ectopic SMAD4 and AP1 site in *Id3*, related to *Figure 5F*.

**Source data 2.** ChIP-qPCR measurement of ectopic SMAD4 and AP1 site in *Wnt7b*, related to *Figure 5G*.

also suggests that without *Prdm16*, the SMAD factors are partly redirected to neighboring genomic regions.

## AP1 is the potential co-factor that interacts with relocated SMAD proteins

We intended to find out how SMAD4 and pSMAD1/5/8 relocate to ectopic sites in the absence of *Prdm16*. By running de novo motif discovery (see Methods), we identified a number of significantly enriched DNA motifs from pSMAD1/5/8 and SMAD4 ChIP-seq peaks in *Prdm16_OE* and *Prdm16_KO* cells (*Figure 5D–E*). Interestingly, in addition to the known SMAD motifs, the only other motif identified in both pSMAD1/5/8 and SMAD4 peaks from *Prdm16_KO* cells but not *Prdm16_OE* cells is the AP1 motif. Furthermore, the AP1 motif is more frequently found in the pSMAD1/5/8 and SMAD4 regions lacking PRDM16 binding than those with PRDM16 binding (*Figure 4—figure supplement 1E*). The interaction between the AP-1 complex and the SMAD proteins was reported previously (*Zhang et al., 1998*). Our result implies that AP-1 is a potential cofactor for the SMAD proteins in the absence of *Prdm16*. To validate this finding, we performed ChIP-qPCR with an antibody against c-FOS, one of the subunits of AP1, on *Prdm16_OE* and *KO* cells. Supporting the global analyses, c-FOS is enriched in the Smad4-bound regions at the *Wnt7b* and *Id3* loci in the *Prdm16_KO* condition (*Figure 5F–G*). There is little c-FOS binding to these regions in cells expressing *Prdm16*, suggesting that there is cooperativity of SMAD and AP-1 which facilitates each other's binding when PRDM16 is absent. Alternatively, PRDM16 may suppress genomic accessibility of these regions for SMAD and AP-1 proteins, a function we reported for PRDM16 in cortical NSCs (*He et al., 2021*).

Together, our genomic data and validation suggest an enhancer switch model (*Figure 5H*): PRDM16 assists SMAD protein binding to repressive cis-regulatory elements that are enriched for GTCTAGAC palindromic motifs; loss of PRDM16 results in genomic relocation of SMAD proteins, presumably via the association with coactivators such as AP1; the consequence of SMAD relocation is de-repression of cell proliferation regulators.

## SMADs and PRDM16 co-repressed genes in NSCs are suppressed in the developing ChP

Next, we investigated whether genes co-regulated by PRDM16 and BMP signaling in NSCs (*Figure 3C*) are also involved in ChP development, where both factors limit NSC proliferation (*Hébert et al., 2002* and *Figure 1*). Using published E12.5 mouse brain single cell RNA-seq (scRNA-seq) data (*La Manno et al., 2021*), we analyzed the expression of the 31 co-repressed genes across ChP epithelial (*Ttr+*), CH (BMP4+/ *Ttr-*) and a few forebrain-specific radial glia (RG) (Sox2+/Hes5+) clusters (*Figure 6A*, *Figure 6—figure supplement 1A*). Except for one gene *Mrtfb* that was not found in the scRNA-seq data, thirteen showed no or weak expression (*Figure 6—figure supplement 1A*), and seventeen had moderate to high expression in at least one cluster (*Figure 6A*). Notably, these genes were generally expressed at lower levels in ChP clusters than in CH or RG clusters, suggesting repression in the ChP.

To test for direct PRDM16 regulation, we performed CUT&TAG assays using the PRDM16 antibody on dissected dorsal midline tissues that mainly contain CH and the ChP (*Figure 6B* and *Figure 6—figure supplement 1B*) and identified 3238 common peaks across three replicates. Manual inspection of the 31 loci revealed PRDM16 binding near the TSS of 25 genes using the browser Embryonic Mouse Brain Epigenome Atlas (*Rhodes et al., 2022*) (Text in black in *Figure 6A*, *Figure 6—figure supplement 1A*). Using an unbiased peak-to-promoter mapping, 24/31 co-repressed and 84/153 BMP-only-repressed genes had PRDM16 binding in E12.5 ChP. The enrichment in the co-repressed

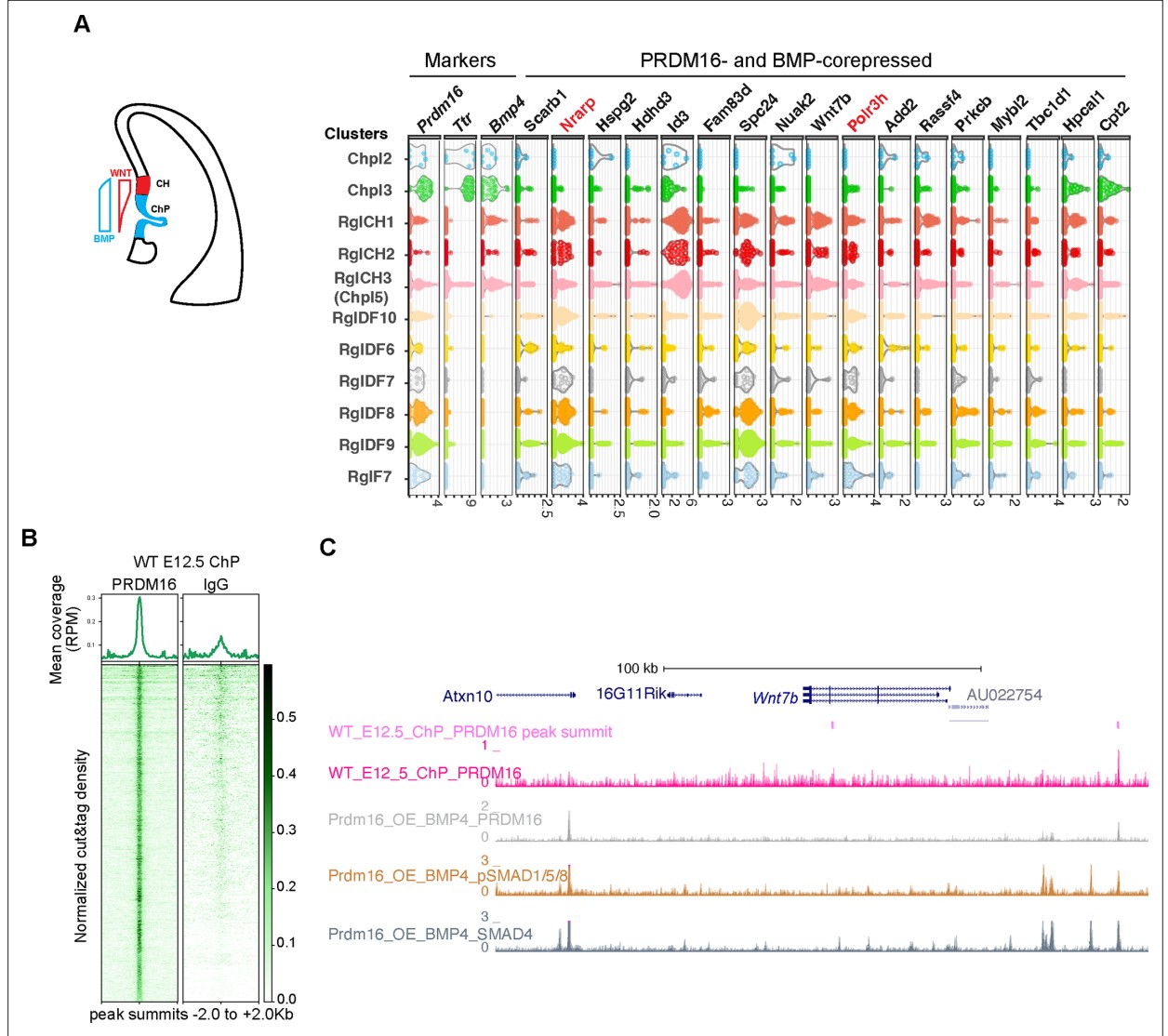

**Figure 6.** PRDM16 and SMADs co-regulated genes exhibit differential expression in choroid plexus (ChP) and neural cells. (**A**) Schematic illustrating the cortical hem (CH) and ChP regions, which exhibit opposing Wnt and BMP signaling gradients. Shown alongside is a violin plot of scRNA-seq expression at E12.5 for the indicated genes across cell type clusters (*Pyrowolakis et al., 2004*). Cell clusters were defined based on marker gene expression from E9.5-E18.5 brain tissues, including choroid plexus clusters 1–5 (Chpl), with Chpl5 newly identified in this study based on *Ttr* expression as well as radial glia clusters: RglCH 1–3, RglDF 6–10, and RglF7. CH: cortical hem; DF: dorsal forebrain; F: forebrain. Gene expression per cell is represented as log2-normalized counts. (**B**) Metaplots and heatmaps showing CUT&TAG signal for PRDM16 and IgG, centered on peak summits and normalized for library size, from E12.5 dorsal midline tissues. (**C**) Genome browser view of the *Wnt7b* locus, displaying CUT&TAG tracks for PRDM16 and ChIP-seq tracks for SMAD proteins.

The online version of this article includes the following figure supplement(s) for figure 6:

**Figure supplement 1.** Wnt genes are potential targets regulated by PRDM16.

group (*Figure 6—figure supplement 1C* Fisher's Exact Test, *P*=0.015) indicates a stronger regulatory role of PRDM16 on these genes in the developing ChP.

As an example, *Wnt7b* has two PRDM16 peaks (*Figure 6C*), one unique to the dorsal midline tissue. Given Wnt signaling's role in ChP specification and NSC proliferation (*Parichha et al., 2022*; *Kalani et al., 2008*). *Wnt7b* likely represents a direct PRDM16 target in the developing ChP.

## PRDM16 represses *Wnt7b* and Wnt activity in the developing ChP

We sought to determine which of the identified genes from NSC culture are regulated by PRDM16 in the developing ChP. In addition to *Wnt7b*, several other Wnt ligands are expressed in the CH and

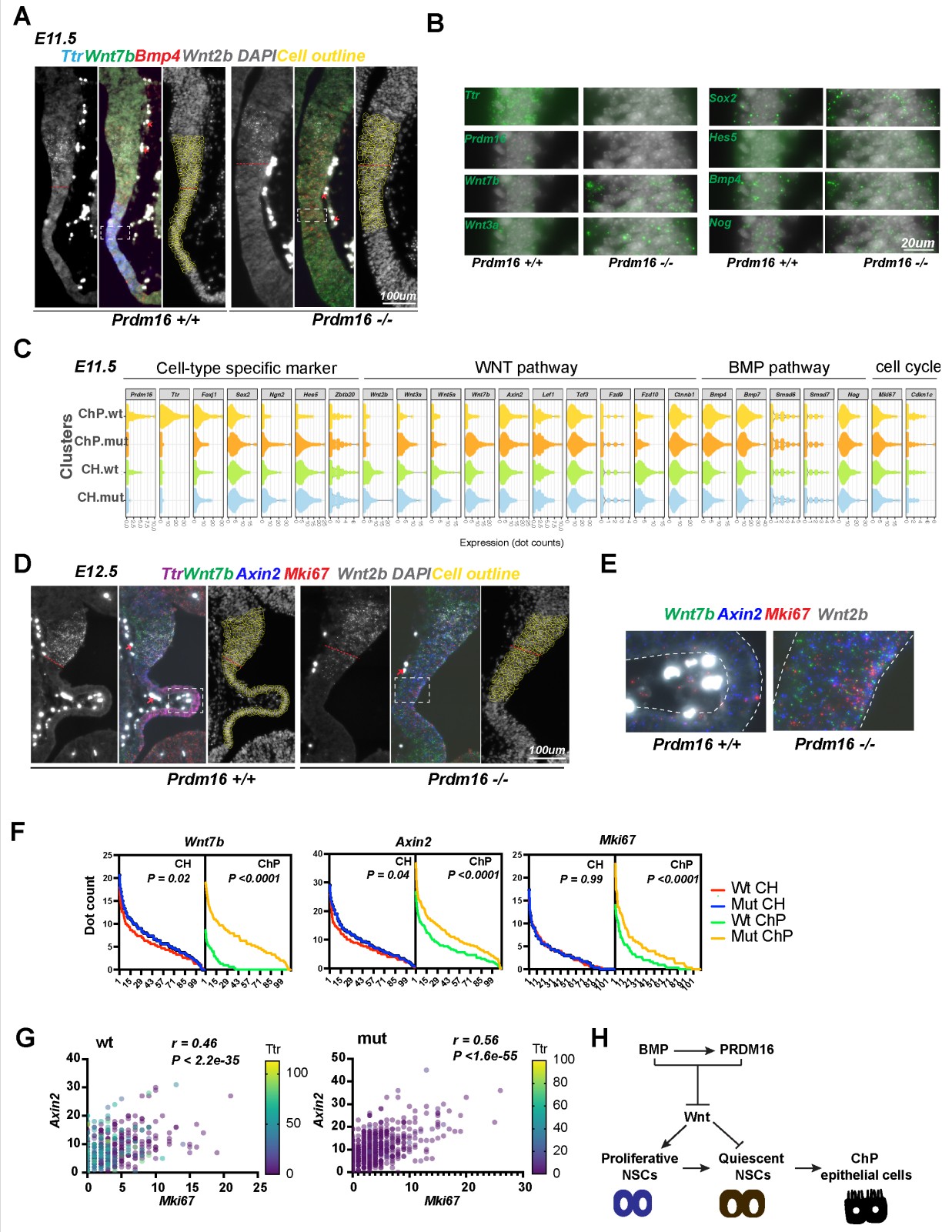

**Figure 7.** PRDM16 suppresses Wnt signaling and cell proliferation in choroid plexus (ChP) epithelial cells. (**A**) SCRINSHOT images of E11.5 wild-type and *Prdm16* mutant brains probed for *Bmp4*, *Ttr*, *Wnt7b* and *Wnt2b*. Red arrows indicate autofluorescent blood cells (non-specific signal in all channels). Cells quantified are circled in yellow, and the red dashed line marks the cortical hem (CH) and ChP boundary. (**B**) Enlarged views of the boxed region in (**A**), showing increased expression of Wnt and neural genes and decreased *Ttr* and *Prdm16* in mutants. (**C**) Violin plots displaying dot counts for the

*Figure 7 continued on next page*

*Figure 7 continued*

24 genes in 330 CH and 330 ChP cells from three wild-type and three *Prdm16* mutant samples. (**D**) SCRINSHOT images of E12.5 wild-type and *Prdm16* mutant brains with probes for *Axin2*, *Ttr*, *Wnt7b*, *Wnt2b*, and *Mki67*. (**E**) Enlarged views of the boxed areas in (**D**), with ChP regions outlined by dashed lines. (**F**) Distribution of dot counts for indicated genes in wild-type (*wt*) and *Prdm16* mutant (*mut*) CH and ChP cells. X-axis: 110 cells per replicate; Y-axis: mean dot count across three biological replicates. P values calculated using area under the ROC curve (Prism Graphpad). (**G**) Multi-variant linear regression of *Mki67*, *Axin2* and *Ttr* expression across 660 CH and ChP cells per genotype. Pearson correlation and significance were calculated using Prism Graphpad. (**H**) Model illustrating the regulatory circuit through which PRDM16 promotes the transition from proliferative to quiescent NSCs, a prerequisite step for ChP epithelial cell specification.

The online version of this article includes the following source data and figure supplement(s) for figure 7:

**Source data 1.** Dot counts in three pairs of *wt* and *Prdm16* mutant CH (1-330) and ChP (331-660) cells from E11.5 brains, related to *Figure 7C*.

**Figure supplement 1.** Full and exemplar segmentation images.

**Figure supplement 2.** Altered expression of Wnt pathway and proliferation genes in *Prdm16* mutant choroid plexus (ChP) cells.

**Figure supplement 2—source data 1.** Dot counts in three pairs of wt and Prdm16 mutant CH (1-330) and ChP (331-660) cells from E12.5 brains, related to *Figure 7—figure supplement 2A*.

**Figure supplement 3.** Expression of Bmp and Wnt genes in the developing fourth ventricle choroid plexus (ChP).

**Figure supplement 4.** Increased expression of additional PRDM16 target genes in *Prdm16* mutant choroid plexus (ChP).

ChP region (*Harrison-Uy and Pleasure, 2012*). For example, *Wnt3a* is expressed in the dorsal-midline region but not in the forebrain, which explains why it was not identified from the forebrain-derived NSC datasets. Although there is no PRDM16 CUT&TAG peak within the *Wnt3a* locus in cultured NSCs, a PRDM16 CUT&TAG peak is present in the intronic region of *Wnt3a* in the dorsal midline tissue (*Figure 6—figure supplement 1D*). We thus examined the scRNA-seq data (*La Manno et al., 2021*) and confirmed that multiple components in the Wnt and BMP pathways are present in the CH and ChP clusters (*Figure 6—figure supplement 1E*). To systematically measure expression changes of PRDM16 target genes in the *Prdm16* mutant brain, we applied a multiplexed fluorescent in situ approach, Single-Cell Resolution in Situ Hybridization On Tissues (SCRINSHOT) (*Sountoulidis et al., 2020*). In this method, each mRNA molecule is hybridized with three gene-specific padlock probes and visualized by fluorescent dye-conjugated detection probes. A dot of fluorescent signal corresponds to one mRNA molecule, allowing quantitative measurement of gene expression in single-cell resolution.

We designed probes for ten Wnt and five BMP pathway components, other genes that are co-repressed by BMP and PRDM16 in NSC culture, as well as cell-type-specific markers (the ChP marker gene *Ttr* and *Foxj1*, the neural progenitor markers *Sox2*, *Hes5*, and *Zbtb20*, and the neuronal marker Ngn2), and conducted SCRINSHOT in three pairs of control and *Prdm16* mutant brains.

In the control brain, the *Ttr* and Foxj1 probes detect the ChP epithelium but not the adjacent tissues at E11.5 and E12.5, confirming the signal specificity of SCRINSHOT. Several Wnt (*Wnt2b*, *Wnt7b*, *Wnt5a*, *Wnt3a*, *Fzd10*, and *Axin2*) and BMP components (*Bmp4* and *Bmp7*) also display tissue specificity: the Wnt genes are more enriched at CH, while the BMP genes are more at the ChP. The representative images from E11.5 are presented in *Figure 7A and B* and *Figure 7—figure supplement 1A*. Except for *Wnt2b*, the other tested *Wnt* genes exhibit low expression in the ChP cells, consistent with the notion that Wnt signaling is present and required at the ChP (*Parichha et al., 2022*).

Next, we quantified expression changes of these genes in *Prdm16* mutant CH and ChP cells. As expression of *Wnt2b* was relatively unchanged in the mutant (*Figure 1—figure supplement 1C*, *Figure 7A*), we used *Wnt2b*-expressing cells to define the CH cells and those expressing *Ttr* to define ChP cells in the wild-type brain slice. There were about 110 cells in each region of one brain slice. Because the mutant brain slice showed little *Ttr* signal, we only used the *Wnt2b*-expressing cells to define the anterior border between CH and neocortex and included 110 cells under the border for the CH cells, and the further 110 cells for the mutant ChP cells, as shown in the images with DAPI and cell outlines (*Figure 7A*, *Figure 7—figure supplement 1B* and *Figure 7D*). We then measured the dot count of each gene in these 220 cells in each sample, and summarized the changes from three pairs of animals in the violin plots (*Figure 7C* and *Figure 7—figure supplement 2A*). Expression of *Ttr* and *Foxj1* is severely reduced, while *Sox2*, *Hes5*, and *Ngn2* all become upregulated in the mutant ChP, indicating a fate transformation of ChP epithelial to neural cells. This agrees with our immunostaining

and conventional in situ results (*Figure 1* and *Figure 1—figure supplement 1*). Similar to *Bmp4*, other BMP components we tested including *Bmp7*, *Smad6*, *Smad7 and Nog* are unchanged, suggesting that PRDM16 is not an upstream regulator of the BMP pathway.

Moreover, *Wnt3a* and *Wnt7b* are significantly upregulated in the *Prdm16* mutant ChP (*Figure 7B-E*, *Figure 7—figure supplement 2A-C*). In line with this, the Wnt target gene *Axin2* is also upregulated, indicating aberrantly elevated Wnt signaling in the *Prdm16* mutant ChP. Thus, the normal function of *Prdm16* represses WNT signaling in the developing ChP.

## Levels of Wnt activity correlate with cell proliferation in the developing ChP and CH

We further assessed the relationship between Wnt signaling and cell proliferation in the ChP and CH at E12.5, by correlating expression levels of *Mki67* with *Wnt3a*, *Wnt7b,* and *Axin2* levels in wild-type and mutant samples. A significant increase in *Mki67* expression in *Prdm16* mutant ChP cells at E12.5 is accompanied by significantly increased Wnt gene expression (*Figure 7F* and *Figure 7—figure supplement 2B–C*). In contrast, little or no increase in *Mki67* signal is found in the mutant CH, suggesting that the effect of PRDM16 is mainly cell-autonomous. We then performed linear regression and Pearson correlation analyses to assess correlations between ChP markers, BMP, and Wnt genes. Interestingly, the level of *Axin2* best correlates with the level of *Mki67*, even better than that with other Wnt genes in both wild-type ($r=0.46$; $p<2.2$ E-35) and mutant ($r=0.56$; $p<1.6$ E-55) CH and ChP cells (*Figure 7G* and *Figure 7—figure supplement 2D*). This result suggests that Wnt activity may be responsible for cell proliferation in the CH and ChP regions. Supporting this finding, it was shown that ectopic Wnt signaling converts ChP cells into proliferative CH neural cells (*Parichha et al., 2022*). Similarly, in the fourth ventricle, two Wnt genes, *Wnt1* and *Wnt3a*, and *Mki67* all became ectopically expressed in *Prdm16* mutant ChP cells (*Figure 7—figure supplement 3*). Thus, PRDM16 also suppresses Wnt signaling in the hindbrain ChP.

Additionally, six cell-proliferation-related genes (*Spc24*, *Spc25*, *Nuf2*, *Ndc80*, *Id3,* and *Mybl2*) exhibited upregulation in *Prdm16* mutant ChP (*Figure 7—figure supplement 4*). Interestingly, all four NDC80 complex genes became upregulated, pointing to its potential role in promoting NSC proliferation. Taken together, our results suggest that specification of the ChP epithelium requires a process transitioning proliferating NSCs into quiescence, and that this process is mediated by a suppressive role of PRDM16 on Wnt signaling and cell cycle regulators.

## Discussion

In this study, we reveal a novel molecular mechanism by which BMP signaling regulates cell proliferation and gene expression. We find that PRDM16 acts as a tethering factor to localize the SMAD4/pSMAD1/5/8 complex at specific genomic sites and facilitate the repressive role of the SMAD proteins. The genes co-repressed by PRDM16 and BMP signaling include those encoding Wnt pathway ligands and other cell proliferation regulators.

Combinatory activities of morphogens are exploited throughout animal development and tissue homeostasis. The crosstalk between BMP and WNT signaling is complex as it can be synergistic or antagonistic (*Itasaki and Hoppler, 2010*). Surprisingly, both types of effects exist in the specification of ChP epithelium. Here, BMP signaling induces NSC quiescence, as evidenced by the presence of ectopic proliferating cells at the ChP in *Bmpr1a* mutant mice (*Hébert et al., 2002*), while Wnt signaling promotes proliferation since the gain-of-function condition of beta-catenin phenocopies *Bmpr1a* mutant animals (*Parichha et al., 2022*). On the other hand, adding a low dose of Wnt activator to cell culture medium enhances programing efficiency of ChP epithelial cells induced by BMP4 (*Pellegrini et al., 2020*), and loss-of-function of beta-catenin resulted in under-developed ChP structure (*Parichha et al., 2022*), suggesting that BMP and WNT signaling collaborate to promote ChP epithelial cell differentiation. Our work demonstrated that PRDM16 ensures the right balance of BMP and Wnt activity by maintaining a low level of Wnt gene expression in the developing ChP.

PRDM16 was shown to physically interact with SMAD3 (*Warner et al., 2007*) and antagonize TGF-β-induced cell cycle arrest (*Takahata et al., 2009*). However, we did not detect stable PRDM16/SMAD4/pSMAD1/5/8 or PRDM16/SMAD4/SMAD3 complexes in co-immunoprecipitation assays (data not shown). Instead, sequential ChIP revealed that PRDM16 and SMAD4 bind to the same DNA

molecule, supporting their cooperative function at co-occupied regulatory elements. Notably, SMADs when associated with PRDM16 compared to those without PRDM16, with SMADs preferentially bind palindromic motifs in the presence of PRDM16 and GC-rich SBE motifs in its absence, suggesting that PRDM16 modulates SMAD DNA-binding specificity and shifts SMAD regulatory output.

A recent study similarly reported that PRDM16 acts as a SMAD4 co-repressor during pancreatic ductal adenocarcinoma progression (*Hurwitz et al., 2023*). There, SMAD4 stably binds the PRDM16 promoter and switches its co-factor from the co-repressor PRDM16 (under low TGF-β) to the co-activator SMAD3 (under high TGF-β), depending on TGF-β levels. This mechanism differs from our model. Our data indicate that high BMP4 levels promote SMAD4 and pSMAD1/5/8 binding to target loci, including *Prdm16*, resulting in both gene repression and activation. Notably, PRDM16 contributes to the repression function of the SMAD complex even under elevated BMP conditions.

Although PRDM16 has been shown to promote proliferation by antagonizing the anti-proliferation activity of TGF-β and SMAD3 (*Takahata et al., 2009*), our data demonstrate that PRDM16 collaborates with SMAD4 and pSMAD1/5/8 to repress proliferation genes, which is consistent with BMP-induced quiescence. Similar roles have been observed in other contexts, such as osteoblast differentiation and maturation (*Wu et al., 2016*). Supporting this, genomic profiling revealed that under high BMP4 and PRDM16 conditions, pSMAD1/5/8 and SMAD4 occupy numerous loci without SMAD3, suggesting PRDM16 primarily modulates BMP target genes in these settings.

Moreover, while it was reported that PRDM16 enhances Wnt signaling by stabilizing nuclear beta-Catenin in the craniofacial tissue (*Shull et al., 2022*), we detected the opposite outcome in the ChP and cultured NSCs, ectopic WNT gene expression in the absence of *Prdm16*, suggesting that PRDM16 represses Wnt genes in these contexts. Notably, PRDM16, Wnt, and BMP co-exist in various developmental settings, such as craniofacial development (*Bjork et al., 2010*; *Shull et al., 2022*; *Nie et al., 2006*), heart formation (*Arndt et al., 2013*; *van Wijk et al., 2007*; *Gessert and Kühl, 2010*), limb patterning (*Bjork et al., 2010*; *Geetha-Loganathan et al., 2008*; *Robert, 2007*), and adult intestinal stem cells (*Stine et al., 2019*; *Korinek et al., 1998*; *He et al., 2004*). We speculate a similar regulatory circuit is used in some, if not all, of these settings.

ChP epithelial cells derive from proliferating NSCs, without intermediate fate-commitment steps. This property endows ChP cells with higher plasticity and makes them more vulnerable to abnormal genetic and cellular change, to which high occurrence frequency of ChP tumors in fetuses and young children may be attributed (*Wolff et al., 2002*). As an essential brain structure, the ChP releases cerebrospinal fluid (CSF) and acts as a brain-blood barrier (*Johansson, 2014*; *Lehtinen et al., 2013*; *Lun et al., 2015*). Its dysfunction has been linked with several types of human diseases including hydrocephalus, Alzheimer's disease, and multiple sclerosis (*Lun et al., 2015*). Deepening our knowledge on the development of ChP will provide new insights into potential therapeutics to prevent or treat ChP tumors and other ChP-related diseases.

Footnote: a study on the antagonism between PRDM16 and SMAD4 was published on Feb 24, 2023 (Hurwitz, 2023) after we posted our manuscript at BioRxiv and during our submission process. Materials and methods.

## Materials and methods

### Animals

All animal procedures were approved by the Swedish Agriculture Board (Jordbruksverket) with document numbers Dnr 11553–2017 and 11766–2022. The *Prdm16*^cGT mice (*Strassman et al., 2017*) were maintained by outcrossing with the FVB/NJ line.

### In situ hybridization

To make probes for conventional RNA in situ hybridization, genomic regions covering one exon or full-length cDNA were PCR amplified to generate fragments with restriction enzyme overhangs. The sequences of all oligos were included in the Key Resources Table. The fragments were inserted into the pBluescript SK(II) vector. In vitro transcription was performed as previously described (*He et al., 2021*). The mouse brains at defined ages were dissected and fixed for 12 hr in 4% PFA, dehydrated in 25% sucrose, cryoprotected, and embedded in O.C.T. The brain samples were then sectioned at 18 μm thickness on Leica cryostats CM3050s. RNA in situ hybridization was performed

using digoxigenin-labeled riboprobes as described previously. Detailed protocols are available upon request. Images were taken using a Leica DMLB microscope.

## Immunostaining

Immunostaining was performed according to standard protocols as previously used (*Dai et al., 2013a*). For EdU and BrdU labeling, EdU (5-ethynyl-2'-deoxyuridine) and BrdU (5-bromo-2'-deoxyuridine) (5–20 µg/g of body weight) were injected into the peritoneal cavity of pregnant mice at desired ages. EdU incorporation was detected with the Click-iT assay (Invitrogen) according to the manufacturer's instructions. BrdU incorporation was measured by immunostaining using an antibody against rat-BrdU (Abcam). Imaging was taken on a Zeiss confocal microscope. ZEN (ZeissLSM800), ImageJ (NIH), and Photoshop (Adobe) were used for analysis and quantification.

## NSC culture

Control and mutant embryonic cortices from E13.5 animals were dissected and dissociated into single-cell suspension and digested with Accutase (Sigma). Cells were maintained in proliferation media (STEMCELL Technologies). Lentivirus expressing Flag-*Prdm16* and a puromycin-resistant gene in the pCDH vector was produced in 293t cells (ATCC) and used to infect a wild-type NSC line derived from E13.5 forebrain, and the selected NSCs were maintained in puromycin-containing medium for three passages before puromycin withdrawal. In the quality control specifications, mycoplasma was not detected.

## RT-qPCR

*Prdm16_OE*, wild-type, and mutant NSCs were cultured and treated with or without BMP4 (Sigma). After 48 hr treatment, total RNAs were extracted using TRIzol reagent (Invitrogen). Total RNA was further cleaned with Turbo DNase (Ambion) and used in reverse transcription with RT master mix (Thermo Fisher). To ensure the absence of genomic DNA, control qPCR was performed on the mock-reverse-transcribed RNA samples. The list of qPCR primers is included in the Key resources table.

## SCRINSHOT

Brain sections were prepared in the same way as for regular in situ hybridization (described above). The SCRINSHOT experiments were carried out according to the published method (*Sountoulidis et al., 2020*). In brief, 3 padlock probes and 3 corresponding detection probes were designed for each gene of interest. The slides with cryo-sectioned brain slices were pretreated at 45 °C to reduce moisture and fixed in 4% PFA in 1 X PBS, followed by washing in PBS Tween-20 0.05% twice. Permeabilization of tissues was done by washing slides in 0.1 M HCl for 2 mins 15 s, followed by washing in PBS Tween-20 0.05% twice. Then a stepwise dehydration was performed for the slides in 70%, 85%, 100% Ethanol and air. The SecureSeal hybridization chamber (GRACE BIO-LABS, 621501) was then mounted to cover each pair of control and *Prdm16* mutant brain slices. Samples were then blocked in a probe-free hybridization reaction mixture of 1X Amplifase buffer (Lucigen, A1905B), 0.05 M KCl, 20% Formamide deionized (Millipore S4117), 0.1 uM Oligo-dT, 0.1 ug/ul BSA (New England Biolabs, B9000S), 1 U/ul RiboLock (Thermo, EO0384), and 0.2 ug/ul tRNAs (Ambion, AM7119). Then hybridization of padlock probes was done by incubating samples with padlock probes (with the concentration of each one 0.01 uM) mixed in blocking reagents used before (no oligo dT used in this step). The slide was then put into a PCR machine to denature at 55 °C for 15 min and hybridize at 45 °C for 120 min. Padlock probes were then ligated by using SplintR ligase (NEB M0375) at 25 °C for 16 hr followed by RCA (rolling cycle amplification) at 30 °C for 16 hr by using phi29 polymerase (Lucigen, 30221–2) and RCA primer1. Then a fixation step was applied to stabilize RCA product in 4% PFA for 15 min followed by washing in PBS Tween-20 0.05%. Then the hybridization of the first 3 genes was done by mixing all 3 3' fluorophore-conjugated detection probes of each gene in reaction reagent 2XSSC, 20% Formamide deionized, 0.1 ug/ul BSA, and 0.5 ng/µl DAPI (Biolegend, 422801) followed by hybridization at 30 °C for 1 hr. The slides were washed in 20% formamide in 2 X SSC and then in 6 X SSC, followed by dehydration in 70%, 85%, and 100% ethanol until the chamber was removed. Then the samples were preserved in SlowFade Gold Antifade mountant (Thermo, S36936) and kept in the dark before imaging. After image acquisition, the first 3 detection probes were removed by using

Uracil-DNA Glycosylase (Thermo, EN0362), and the slides were ready for the next round of detection probe hybridization. The procedure was repeated until all genes were hybridized and imaged.

Images were acquired with the Zeiss Axio Observer 7 fluorescent microscope with an automated stage setting to fix the imaging region for different hybridization rounds. Image analysis was done according to the published method (*Sountoulidis et al., 2020*). In brief, the DAPI channel from each round was extracted to measure the shift of imaging. The images were then aligned for all gene channels in Zen by creating an image subset with the shifting value. After alignment, images were exported into TIFF files and the threshold analysis was carried out for individual channels one by one in CellProfiler with scripts provided by the published method (*Sountoulidis et al., 2020*). Then, nuclear segmentation was done manually in the Fiji ROI manager to obtain nuclear ROIs, followed by an expansion of 2 μm in CellProfiler to obtain cell ROIs. Then signal dots were counted in these cell ROIs for each gene in CellProfiler and Fiji. Summary of the dot counts for each gene was exported to Excel files.

## CUT&TAG

CUT&TAG was performed according to the published method (*Kaya-Okur et al., 2019*). In brief, *Prdm16_OE* and *Prdm16* mutant NSCs were cultured in the NeuroCult Proliferation Kit (Stem Cell Technologies, 05702) for 3 days with or without BMP4 at concentration of 25 ng/ml medium (Sigma, H4916). Cells on day 3 were then shortly rinsed with 1 X PBS and resuspended in 1 ml cold NE1 buffer on ice. Nuclei were lightly fixed with 0.1% formaldehyde to 1 ml PBS at RT for 2 min and neutralized with 75 mM Glycine. Nuclei were then washed three times, resuspended in 1 mL wash buffer and mixed with 90 μL Concanavalin A-coated magnetic beads (Polyscience 86057). The nuclei/beads mix was then blocked with 800 μL cold Antibody buffer for 5 min and resuspended in 1.2 mL Antibody buffer and aliquoted into 8 tubes with 150 μL each. 2 μL Anti-PRDM16 (Generous gift from Bryan Bjork lab) or IgG (SIGMA-ALDRICH, I5006) was added into each tube for an overnight incubation at 4 °C. The beads/nuclei/antibody mix was then washed with Dig-wash buffer and incubated with secondary antibody (1:100 dilution) for 1 hr. After further washes with the Dig-wash buffer. a pA-Tn5 adaptor complex in Dig-300 buffer was added to the beads for 1 hr reaction. The beads were then washed with Dig-300 buffer before the incubation with 300 μl Tagmentation buffer at 37 °C for 1 hr. Then 10 μl 0.5 M EDTA, 3 μL 10% SDS and 2.5 μL 20 mg/mL proteinase K (Invitrogen) were used to stop the reaction. Then the resultant DNA was purified with the DNA Clean & Concentrator kit (ZYMO Research D4013) and eluted in 25 uL Elution buffer. To generate libraries, 21 μL fragment DNA was mixed with 2 μL 10 uM Universal i5 primer, 2 μL 10 μM uniquely barcoded i7 primer and 25 μL PCR master mix (NEB Phusion High-Fidelity PCR Master Mix with HF Buffer, M0531). The PCR condition was as follows: 72 °C 5 min, 98 °C 30 s, repeat 12 times (98 °C 10 s, 63 °C 10 s), 72 °C 1 min and hold 4 °C. The libraries were cleaned with standard Ampure XP beads as previously described. Libraries from four biological replicates were produced for each condition.

## ChIP

ChIP was performed as previously described (*Dai et al., 2013b*). For each ChIP reaction, 10 million *Prdm16_OE*, Prdm16_KO NSCs with or without 3 day BMP4 treatment were fixed, lysed, sonicated and made into chromatin extract. After precleared with gamma-bind-G beads, the chromatin extract was incubated with 2 μg PRDM16, 2 μg SMAD4 (Proteintech, 10231–1-AP), 2 μg pSMAD1/5/8 (Millipore, AB3848-1), or 2 μg SMAD3 (abcam, ab227223) in each ChIP reaction. In sequential ChIP assays, the same chromatin lysate was precleared and used, but with more antibodies in each reaction: 5 μg of IgG or PRDM16 antibody in the first round of immunoprecipitation. The elutes were then precleared again using gamma-bind G beads, divided into two equal halves and immunoprecipitated with 2 μg of IgG or the SMAD4 antibody. In ChIP-qPCR experiments, 2 μg SMAD4 and 2 μg c-FOS antibodies (Invitrogen, MA5-15055) were used in each reaction. The precipitated DNA was reverse cross-linked, and then purified using the Qiagen PCR purification kit.

## Computation analyses
### ChIP-seq libraries and analyses

0.2% input and ChIPed DNA were made into libraries using the NEBNext Ultra II DNA Library Prep Kit and sequenced on the Illumina Nextseq500 platform. Three replicates of ChIP-seq samples, after

the adaptor trimming by Trimmomatic, were mapped to the UCSC *Mus musculus* (mm10) genome assembly using Bowtie2 with the default parameters. The uniquely mapped reads (with mapping quality ≥ 20) were used for further analyses. The peaks were called by HOMER (v4.10) (*Heinz et al., 2010*). The reproducibility between replicates was estimated by Irreproducibility Discovery Rate (IDR), using the HOMER IDR pipeline (https://github.com/karmel/homer-idr; *Allison, 2015*). As suggested by the Encode IDR guideline, we used a relatively relaxed parameter '-F 2 -fdr 0.3 P.1 L 3 -LP.1' for the true/pseudo/pooled replicates by the HOMER peak calling. The final confident peaks were determined by an IDR <5%. The peaks that were overlapped with the mm10 blacklist were also removed.

## CUT&TAG analyses

The CUT&TAG samples of *Prdm16_OE* and *Prdm16* mutant NSCs in four replicates, and of Prdm16 from CH and ChP area at stage 12.5 embryos in three replicates, were mapped to mm10 genome assembly using Bowtie2 (bowtie2 --end-to-end --very-sensitive --no-mixed --no-discordant --phred33 -I 10 X 700). The CUT&TAG coverage for these samples was generated by bedtools genomeCoverageBed and normalized to the library depths to give a read per million per base (RPM). By using the peak caller SEACR (*Meers et al., 2019*), which was designed for calling peaks from sparse chromatin profiling data such as CUT&TAG [22], we performed peak calling with FDR <10% for each replicated sample (SEACR_1.3.sh normalized_coverage 0.1 non-stringent output_peaks). To identify confident peaks, we selected the common peaks, which were called from all replicates for each condition. Peak overlapping analysis was performed by the Homer mergePeaks function with default parameters. Peak overlap analysis utilized HOMER's mergePeaks function, specifically employing the default '-d given' parameter. This approach ensures that only peaks directly overlapping with each other are merged, maintaining the precise distances between peaks as provided in the input peak calls. This method is particularly beneficial for retaining the integrity of the original peak boundaries, avoiding any alterations to their relative positions. De novo motif discovery from SMAD4 and pSMAD1/5/8 peaks in *Prdm16_OE* NSCs and Prdm16 mutant NSCs was performed by MEME-ChIP software (*Machanick and Bailey, 2011*) (parameters: -ccut 100 -meme-p 5 -dna -meme-mod anr -minw 5 -maxw 15 -filter-thresh 0.05).

The CUT&TAG samples of H3K4me3 BMP4-treated and non-treated *Prdm16_OE* and *Prdm16_KO* cells in two or three replicates were mapped to the mm10 genome assembly using Bowtie2 (bowtie2 --end-to-end --very-sensitive --no-mixed --no-discordant --phred33 -I 10 X 700). Differential analysis of TSS up-stream and down-stream 500 bp between *Prdm16_OE* BMP-treated vs non-BMP-treated and Prdm16_KO vs *Prdm16_OE* BMP-treated was performed by the Limma R package (*Ritchie et al., 2015*). Gene Ontology (GO) enrichment analysis of up-regulated and down-regulated regions was performed by PANTHER (*Thomas et al., 2022*).

## scRNA-seq analyses

The scRNA-seq data of the developing mouse brain were obtained from *La Manno et al., 2021*. The counts per cell were normalized by the logNormCounts function in the Bioconductor package 'scater' and the normalized expression data per cell were used to generate gene expression violin plots. To test whether Prdm16 target gene sets of Prdm16 with and without BMP4 and ChP E12.5 are significantly enriched among the scRNA-seq gene mean expression in each identified cell type cluster, the gene set enrichment analysis (GESA) 'fgesa' Bioconductor package was used (*Korotkevich et al., 2021*).

## Acknowledgements

We thank the animal experimental core facility and the imaging facility of Stockholm University and Bioinformatics and Expression Analysis Core Facility at Karolinska Institute, Sweden, for their service and support. We thank Christos Samakovlis and his lab members for technical help on SCRINSHOT experiments. We also appreciate technical help from Adrian Martinez Martin. JW is funded by the Australian Research Council Centre of Excellence for the Mathematical Analysis of Cellular Systems (CE230100001) and ANU Future Scheme. The project was supported by the research project grant from Swedish Research Council (Vetenskapsrådet, 2020–03543) and the research grants from Swedish Cancerfonden (CAN 2017/529 and 20 1046 PjF 01 H) to QD.

## Additional information

### Funding

| Funder | Grant reference number | Author |
|---|---|---|
| Vetenskapsrådet | 2020-03543 | Qi Dai |
| Cancerfonden | CAN 2017/529 | Qi Dai |
| Cancerfonden | 20 1046 PjF 01 H | Qi Dai |
| Australian Research Council Centre of Excellence for the Mathematical Analysis of Cellular Systems | CE230100001 | Jiayu Wen |

The funders had no role in study design, data collection and interpretation, or the decision to submit the work for publication.

### Author contributions

Li He, Data curation, Formal analysis, Validation, Investigation, Visualization, Methodology, Writing – review and editing; Jiayu Wen, Software, Formal analysis, Validation, Methodology; Qi Dai, Conceptualization, Resources, Data curation, Supervision, Funding acquisition, Investigation, Writing – original draft, Project administration, Writing – review and editing

### Author ORCIDs

Li He ⓘD https://orcid.org/0000-0002-4743-5410
Jiayu Wen ⓘD https://orcid.org/0000-0003-1249-6456
Qi Dai ⓘD https://orcid.org/0000-0002-2082-0693

### Ethics

All animal work was performed in strict accordance with the ethic permit approved by Swedish Board of Agriculture (Dnr 11553-2017 and 11766-2022).

Reviewer #1 (Public review): https://doi.org/10.7554/eLife.104076.3.sa1
Reviewer #2 (Public review): https://doi.org/10.7554/eLife.104076.3.sa2
Reviewer #3 (Public review): https://doi.org/10.7554/eLife.104076.3.sa3
Author response https://doi.org/10.7554/eLife.104076.3.sa4

## Additional files

### Supplementary files

MDAR checklist

### Data availability

All sequencing data produced in this study has been submitted to Gene Expression Omnibus (GEO accession number: GSE275758).

The following dataset was generated:

| Author(s) | Year | Dataset title | Dataset URL | Database and Identifier |
|---|---|---|---|---|
| He L, Wen J, Dai Q | 2025 | PRDM16 functions as a co-repressor in the BMP pathway to suppress neural stem cell proliferation | https://www.ncbi.nlm.nih.gov/geo/query/acc.cgi?acc=GSE275758 | NCBI Gene Expression Omnibus, GSE275758 |

The following previously published dataset was used:

| Author(s) | Year | Dataset title | Dataset URL | Database and Identifier |
|---|---|---|---|---|
| La Manna G, Siletti K, Furlan A | 2021 | Single cell sequencing of the development mouse brain | https://www.ncbi.nlm.nih.gov/sra?term=PRJNA637987 | NCBI Sequence Read Archive, PRJNA637987 |

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

# Appendix 1

**Appendix 1—key resources table**

| Reagent type (species) or resource | Designation | Source or reference | Identifiers | Additional information |
|---|---|---|---|---|
| Genetic reagent (*Mus musculus*) | *FVB/NJ* | Strassman, A., et al., 2017 | | A gift from Bryan C. Bjork's laboratory |
| Genetic reagent (*Mus musculus*) | *Prdm16* GT | Strassman, A., et al., 2017 | | A gift from Bryan C. Bjork's laboratory |
| Antibody | Rabbit polyclonal anti-PRDM16 | Gift from Bryan Bjork's lab | | IF (1:500–1000), WB (1:2000) Cut&Tag (1:150) |
| Antibody | Goat polyclonal anti-SOX2 | R&D systems | AF2018 | IF (1:1000) |
| Antibody | Rabbit polyclonal anti-Ki67 | Abcam | ab15580 | IF (1:1000) |
| Antibody | Goat polyclonal anti-Doublecortin | SANTA CRUZ Biotechnology | sc-8066 | IF (1:500) |
| Antibody | beta Galactosidase Polyclonal Antibody | Thermo Fisher | A-11132 | IF (1:500) |
| Antibody | Rabbit polyclonal anti-SMAD4 | Proteintech | 10231–1-AP | IF (1:500) WB (1:2000) ChIP (1:150) |
| Antibody | Rabbit polyclonal anti-pSMAD1/5/8 | Millipore | AB3848-1 | IF (1:500) WB (1:2000) ChIP (1:150) |
| Antibody | Secondary antibodies conjugated to Cy3 or Cy5 or Alexa Fluor-488 | Jackson ImmunoResearch | | IF (1:1000) |
| Antibody | mouse anti-Rabbit IgG light chain specific | Jackson ImmunoResearch | Cat# 211-032-171 | WB (1:40000) |
| Antibody | Rabbit polyclonal anti-SMAD3 | abcam | ab227223 | IF (1:500) WB (1:1000) ChIP (1:150) |
| Antibody | Rabbit monoclonal anti-C-Fos | Invitrogen | MA5-15055 | ChIP (1:150) |
| Antibody | Rabbit polyclonal to Histone H3 (tri methyl K4) | abcam | ab8580 | CUT&TAG (1:150) |
| Antibody | IgG from rabbit serum | SIGMA-ALDRICH | I5006 | ChIP (1:150) CUT&TAG (1:150) |
| Antibody | Guinea Pig anti-Rabbit IgG (Heavy & Light Chain) Antibody - Preadsorbed | antibodies-online | ABIN101961 | ChIP (1:150) CUT&TAG (1:150) |
| Cell line | 293t | ATCC | CRL-3216 | |
| Sequence-based reagent | Ttr ISH F | This paper | PCR primers | GCTCTAGAGGATGGCTTCCCTTCGAC |
| Sequence-based reagent | Ttr ISH R | This paper | PCR primers | CCCAAGCTTAAAAATGCTTCCCGGCAT |
| Sequence-based reagent | Wnt2b ISH F | This paper | PCR primers | GCTCTAGAGAAGCTGCAGGGTGAGGAT |
| Sequence-based reagent | Wnt2b ISH R | This paper | PCR primers | GGAATTCTCATATCGCCTCCTCAGGTAGT |
| Sequence-based reagent | Bmp4 ISH F | This paper | PCR primers | CGGGATCCCCTGCAGCGATCCAGTCT |
| Sequence-based reagent | Bmp4 ISH R | This paper | PCR primers | GGAATTCGCCCAATCTCCACTCCCT |
| Sequence-based reagent | 18srRNA qRTF | This paper | RTqPCR primers | GTAACCCGTTGAACCCCATT |

*Appendix 1 Continued on next page*

*Appendix 1 Continued*

| Reagent type (species) or resource | Designation | Source or reference | Identifiers | Additional information |
|---|---|---|---|---|
| Sequence-based reagent | 18srRNA qRTR | This paper | RTqPCR primers | CCATCCAATCGGTAGTAGCG |
| Sequence-based reagent | Mki67 qRTF1F | This paper | RTqPCR primers | TGCCTCAGATGGCTCAAAGA |
| Sequence-based reagent | Mki67 qRTF1R | This paper | RTqPCR primers | TCTGCCAGTGTGCTGTTCTA |
| Sequence-based reagent | Wnt7b qRTF | This paper | RTqPCR primers | TGCCTTCACCTATGCCATCA |
| Sequence-based reagent | Wnt7b qRTR | This paper | RTqPCR primers | GTCACAGCCACAATTGCTCA |
| Sequence-based reagent | Prdm16 qRTF | This paper | RTqPCR primers | CACCGGGTCAGAGGAGAAAT |
| Sequence-based reagent | Prdm16 qRTR | This paper | RTqPCR primers | CGACATGTCAGGGCTCCTAT |
| Sequence-based reagent | Cdkn1a qRTF | This paper | RTqPCR primers | CAAAGTGTGCCGTTGTCTCT |
| Sequence-based reagent | Cdkn1a qRTR | This paper | RTqPCR primers | CGAAGTCAAAGTTCCACCGT |
| Sequence-based reagent | Mybl2 qRTF | This paper | RTqPCR primers | CCTGGAGCAGAGAGACAACA |
| Sequence-based reagent | Mybl2 qRTR | This paper | RTqPCR primers | GTACTGGCATTGCTGGTCTG |
| Sequence-based reagent | Ndc80 qRTF | This paper | RTqPCR primers | CGCCTCTCTATGCAGGAGTT |
| Sequence-based reagent | Ndc80 qRTR | This paper | RTqPCR primers | TTCCGATGTCGGTTTGTGTG |
| Sequence-based reagent | Nuf2 qRTF | This paper | RTqPCR primers | GAACAAACCGGACAAGTCGT |
| Sequence-based reagent | Nuf2 qRTR | This paper | RTqPCR primers | CTTCCTGGTGCACATTGCTT |
| Sequence-based reagent | Spc24 qRTF | This paper | RTqPCR primers | TGCTTTGAGACAGCATGGAG |
| Sequence-based reagent | Spc24 qRTR | This paper | RTqPCR primers | AGCCCTCTCTAACCTCTCCT |
| Sequence-based reagent | Spc25 qRTF | This paper | RTqPCR primers | AGAGTCGGAAGAGCTGACTG |
| Sequence-based reagent | Spc25 qRTR | This paper | RTqPCR primers | CGCTGATTTCTGCAGTCCTT |
| Sequence-based reagent | Wnt7b Co-bound enhancer ChIP qPCRF | This paper | ChIPqPCR primers | GTCATGTGCTTTCAGGGCAT |
| Sequence-based reagent | Wnt7b Co-bound enhancer ChIP qPCRR | This paper | ChIPqPCR primers | TCCTAGGAGTCACCATGGAA |
| Sequence-based reagent | Id1 Co-bound enhancer ChIP qPCRF | This paper | ChIPqPCR primers | GAGGCCACTCCAGTTCCAAT |
| Sequence-based reagent | Id1 Co-bound enhancer ChIP qPCRR | This paper | ChIPqPCR primers | GCGGTTCACATCTGGAGTCT |
| Sequence-based reagent | Wnt7b ectopic enhancer ChIP qPCRF | This paper | ChIPqPCR primers | CAGCGGTCAAGTTTAAGCGA |
| Sequence-based reagent | Wnt7b ectopic enhancer ChIP qPCRR | This paper | ChIPqPCR primers | CCACAGATGTCCCAGAGTGA |
| Sequence-based reagent | Id3 ectopic enhancer ChIP qPCRF | This paper | ChIPqPCR primers | CTTGGGCAATGTGAAAGGCT |
| Sequence-based reagent | Id3 ectopic enhancer ChIP qPCRR | This paper | ChIPqPCR primers | CAGTTGGTCCCTGTCAGTGT |
| Sequence-based reagent | Prdm16_PL_4 | This paper | SCRINSHOT padlock probe | /5Phos/AAGTCTTCAGAGATCTGCTTTTCCTCT ATGATTACTGACTGCGTCTATTTAGTGGAGCCG CCCCTATCTTCTTTCAGAACTTCTCGCTACCC |
| Sequence-based reagent | Ttr_Pl-1 | This paper | SCRINSHOT padlock probe | /5Phos/CTTCTACAAACTTCTCATCTGTCCTCT ATGATTACTGACTGCGTCTATTTAGTGGAGCC GCTACTATCTTCTTTCAGTTCTACTCTGTACACTC |
| Sequence-based reagent | Foxj1_PL_1 | This paper | SCRINSHOT padlock probe | CGTGATCCACTTGTAGATGGTCCTCTATGATTA CTGACTGCGTCTATTTAGTGGAGCCGCCCCTAT CTTCTTTCGGAAGTAGCAGAAGTTGTC |

*Appendix 1 Continued on next page*

Appendix 1 Continued

| Reagent type (species) or resource | Designation | Source or reference | Identifiers | Additional information |
|---|---|---|---|---|
| Sequence-based reagent | Foxj1_PL_2 | This paper | SCRINSHOT padlock probe | /5Phos/GCCAAGAAGGTCTCATCAAAGTCCTCTA TGATTACTGACTGCGTCTATTTAGTGGAGCCGC CCCTATCTTCTTTGATGCTGTAGGAAGGATGTG |
| Sequence-based reagent | Foxj1_PL_3 | This paper | SCRINSHOT padlock probe | /5Phos/CACTTTGATGAAGCACTTGTTCTCCTCTA TGATTACTGACTGCGTCTATTTAGTGGAGCCGCCCC TATCTTCTTTCTCATCCTTCTCCCGAGG |
| Sequence-based reagent | Sox2_Pl-2 | This paper | SCRINSHOT padlock probe | /5Phos/TCGGCAGCCTGATTCCTCCTCTATGATTA CTGACTGCGTCTATTTAGTGGAGCCACGTCTATCTT CT TTTAGTTCAAATATATACATGGATTC |
| Sequence-based reagent | Sox2_Pl-3 | This paper | SCRINSHOT padlock probe | /5Phos/CTCTAAATATTACTGGTAAAGACTCCTCTATG AT TACTGACTGCGTCTATTTAGTGGAGCCACGTCTATC TT CTTTCATCGCCCGGAGTCTAG |
| Sequence-based reagent | Ngn2_Pl-1 | This paper | SCRINSHOT padlock probe | /5Phos/TTGGTTTGACAGGTGAAATTCTCCTCTATGAT T ACTGACTGCGTCTATTTAGTGGAGCCATCGCTATCT TC TTTCACATCAGAGAGGGAAAGT |
| Sequence-based reagent | Ngn2_Pl-2 | This paper | SCRINSHOT padlock probe | /5Phos/ACGCTTGCATTCAACCCTTTCCTCTATGATTA C TGACTGCGTCTATTTAGTGGAGCCATCGCTATCTTC TT TGAGAATTCAGCCTAAATTTCC |
| Sequence-based reagent | Ngn2_Pl-3 | This paper | SCRINSHOT padlock probe | /5Phos/ATAAACCAGAGCTGGTCTCTCCTCTATG ATTA CTGACTGCGTCTATTTAGTGGAGCCATCGCTATCTT C TTTCCATAAATCCTCACATCTTCA |
| Sequence-based reagent | Hes5_Pl-1 | This paper | SCRINSHOT padlock probe | /5Phos/CAAGAGTTCAAATTCTAAGTACTCCTCTATGA T TACTGACTGCGTCTATTTAGTGGAGCCAGCTCTATC TTC TTTTGTGTTCTCCCACATGAC |
| Sequence-based reagent | Hes5_Pl-2 | This paper | SCRINSHOT padlock probe | /5Phos/CACACATTCTCTAAGAATGACTCCTCTATGAT TACTGACTGCGTCTATTTAGTGGAGCCAGCTCTATCT TCTTTCCCAAATGACAACTCTGCA |
| Sequence-based reagent | Hes5_Pl-3 | This paper | SCRINSHOT padlock probe | /5Phos/TTAAGGATCATCGTGGAGACTCCTCTAT GATT ACTGACTGCGTCTATTTAGTGGAGCCAGCTCTATCT T CTTTCACCCATACAAAGGAATCCT |
| Sequence-based reagent | Zbtb20_PL-1 | This paper | SCRINSHOT padlock probe | /5Phos/CCATTTCCTGCTTGATGTGTCCTCTATGATTA CTGACTGCGTCTATTTAGTGGAGCCGCCCCTATCTT C TTTCATAGTAGTCATAGTCATCTT |
| Sequence-based reagent | Zbtb20_PL-2 | This paper | SCRINSHOT padlock probe | /5Phos/TACTTATCCGTCAGACACATTCCTCTATGATT A CTGACTGCGTCTATTTAGTGGAGCCGCCCCTATCTT C TTTGTAAGAAAGAGAGAAAGGTAC |
| Sequence-based reagent | Zbtb20_PL-3 | This paper | SCRINSHOT padlock probe | /5Phos/GAATGTGTCGAGTGGATGAGTCCTCTAT GATT ACTGACTGCGTCTATTTAGTGGAGCCGCCCCTATCT T CTTTGAGCGTGAGAGTTTGTCAGT |

Appendix 1 Continued on next page

*Appendix 1 Continued*

| Reagent type (species) or resource | Designation | Source or reference | Identifiers | Additional information |
|---|---|---|---|---|
| Sequence-based reagent | Wnt2b_Pl-1 | This paper | SCRINSHOT padlock probe | /5Phos/TGAAGACAGACCTCTCCTATCCTCTATGATTACTGACTGCGTCTATTTAGTGGAGCCCGTACTATCTTC TTTTTAGTGCAACTCAAATCACTC |
| Sequence-based reagent | Wnt2b_Pl-2 | This paper | SCRINSHOT padlock probe | /5Phos/GTAATGGATGTTGTCACTACATCCTCTATGATT ACTGACTGCGTCTATTTAGTGGAGCCCGTACTATCTT CTTTCTTGGCAAAGCGAACACC |
| Sequence-based reagent | Wnt2b_Pl-3 | This paper | SCRINSHOT padlock probe | /5Phos/CGTGCAGTAGTTAACAGTTGTCCTCTAT GATT ACTGACTGCGTCTATTTAGTGGAGCCCGTACTATCTT CTTTCTGTTTAAGTCGATCGTGGAA |
| Sequence-based reagent | Wnt3a_Pl-1 | This paper | SCRINSHOT padlock probe | /5Phos/CCTCTCTACTCAGGATAGAGTCCTCTATGAT TACTGACTGCGTCTATTTAGTGGAGCCGCCCCTATC TTCTTTCCATAGAGATTCCTAGAGAAC |
| Sequence-based reagent | Wnt3a_Pl-2 | This paper | SCRINSHOT padlock probe | /5Phos/ATGAGACCCTCACAGGTAGGTCCTCTATGAT TACTGACTGCGTCTATTTAGTGGAGCCGCCCCTATC TTCTTTCAGAAACCGGGTCCTTAGGT |
| Sequence-based reagent | Wnt3a_Pl-3 | This paper | SCRINSHOT padlock probe | /5Phos/CATGTATCTAGATAGAGCAGTGTCCTCTAT GATTACTGACTGCGTCTATTTAGTGGAGCCGCCCC TATCTTCTTTCACCCTGAAGCACCCTCT |
| Sequence-based reagent | Wnt5a_PL-1 | This paper | SCRINSHOT padlock probe | /5Phos/TTAAGAGCCACAGGACTGATCCTCTATGAT TACTGACTGCGTCTATTTAGTGGAGCCGCCCCTAT CTTCTTTAATGAATATTGTTGGGCAATAAG |
| Sequence-based reagent | Wnt5a_PL-2 | This paper | SCRINSHOT padlock probe | /5Phos/TAGAGACCACCAAGAATTAGTCCTCTAT GATT ACTGACTGCGTCTATTTAGTGGAGCCGCCCCTATCT TC TTTTGAACAGGGTTATTCATACC |
| Sequence-based reagent | Wnt5a_PL-3 | This paper | SCRINSHOT padlock probe | /5Phos/CTTCATTGTTGTGTAAGTTCATGTCCTCTATG A TTACTGACTGCGTCTATTTAGTGGAGCCGCCCCTAT CT TCTTTATACTGTCCTACGGCCTG |
| Sequence-based reagent | Wnt7b_PL-1 | This paper | SCRINSHOT padlock probe | /5Phos/AGGGATTTGTCTCCATTGGTCCTCTATGATTA C TGACTGCGTCTATTTAGTGGAGCCGCCCCTATCTTC T TTCACTTTCCCAAAGAGAAGTAA |
| Sequence-based reagent | Wnt7b_PL-2 | This paper | SCRINSHOT padlock probe | /5Phos/CATTGTTGTGAAGGTTCATGAGTCCTCT ATGATT ACTGACTGCGTCTATTTAGTGGAGCCGCCCCTATCT TCT TTGAACCTTTCTGCCCGCCT |
| Sequence-based reagent | Wnt7b_PL-3 | This paper | SCRINSHOT padlock probe | /5Phos/CCGGTCACAGCCACAATTGTCCTCTATG ATTACT GACTGCGTCTATTTAGTGGAGCCGCCCCTATCTTCT TT GTTGTAGTAGCCTTGCTTCTC |
| Sequence-based reagent | Axin2_PL-1 | This paper | SCRINSHOT padlock probe | /5Phos/AAGTCAAGAACACCTGGTAGTCCTCTAT GATTA CTGACTGCGTCTATTTAGTGGAGCCGCCCCTATCTT C TTTCATATTCCAGGTAAATGTCAG |

*Appendix 1 Continued on next page*

*Appendix 1 Continued*

| Reagent type (species) or resource | Designation | Source or reference | Identifiers | Additional information |
|---|---|---|---|---|
| Sequence-based reagent | Axin2_PL-2 | This paper | SCRINSHOT padlock probe | /5Phos/TACGTGATAAGGATTGACTGGTCCTCTATGATT ACTGACTGCGTCTATTTAGTGGAGCCGCCCCTATCT CTTTCAAAGACATAGCCGGAACC |
| Sequence-based reagent | Axin2_PL-3 | This paper | SCRINSHOT padlock probe | /5Phos/TCCTGTATGGAATTTCTTCTCTCCTCTATGAT TACTGACTGCGTCTATTTAGTGGAGCCGCCCCTATC TTCTTTCTTTGAGCCTTCAGCATCC |
| Sequence-based reagent | Lef1_Pl-1 | This paper | SCRINSHOT padlock probe | /5Phos/CATATTGGGCATCATTATGTAGTCCTCTATGA TTACTGACTGCGTCTATTTAGTGGAGCCACTGCTAT CTTCTTTCATGTACGGGTCGCTGTT |
| Sequence-based reagent | Lef1_Pl-2 | This paper | SCRINSHOT padlock probe | /5Phos/CGTGCTAGTTCATAGTATTTGTCCTCTATGA TTACTGACTGCGTCTATTTAGTGGAGCCACTGCT ATCTTCTTTGTGTAGCTGTCTCTCTTTC |
| Sequence-based reagent | Lef1_Pl-3 | This paper | SCRINSHOT padlock probe | /5Phos/CTTCATCAGACACTTCACAGTCCTCTATG ATTACTGACTGCGTCTATTTAGTGGAGCCACTGC TATCTTCTTTGGGTTTGGCTATCTCTTTAG |
| Sequence-based reagent | Tcf3_PL_1 | This paper | SCRINSHOT padlock probe | /5Phos/AGGTCACTCAGTTCCTTGTCTCCTCTATG ATTACTGACTGCGTCTATTTAGTGGAGCCGCCCC TATCTTCTTTACATCATGCTGAAGTCCAGG |
| Sequence-based reagent | Tcf3_PL_2 | This paper | SCRINSHOT padlock probe | /5Phos/GAGACCTGTCTCATCCAGAATCCTCTATGAT TACTGACTGCGTCTATTTAGTGGAGCCGCCCCTAT CTTCTTTTCTCTGCTCATGCTTCGATG |
| Sequence-based reagent | Tcf3_PL_3 | This paper | SCRINSHOT padlock probe | /5Phos/TCTTTCTCCTCCCGCTTGATTCCTCTATGAT TACTGACTGCGTCTATTTAGTGGAGCCGCCCCTATC TTCTTTCTGATGCGATTTCCTCATCC |
| Sequence-based reagent | Fzd9-PL-1 | This paper | SCRINSHOT padlock probe | /5Phos/CACATAGAAAGGAAGATAATCTCCTCTA TGATTACTGACTGCGTCTATTTAGTGGAGCCGCC CCTATCTTCTTTCAAGGAGTAGACATTGTAG |
| Sequence-based reagent | Fzd9-PL-2 | This paper | SCRINSHOT padlock probe | /5Phos/TTCATAAACGTAGCAGACAATGTCCTCTA TGATTACTGACTGCGTCTATTTAGTGGAGCCGCC CCTATCTTCTTTGAAGTCCATGTTGAGGCG |
| Sequence-based reagent | Fzd9-PL-3 | This paper | SCRINSHOT padlock probe | /5Phos/CACTGGACCCTGCCCTCCTCTATGATTAC TGACTGCGTCTATTTAGTGGAGCCGCCCCTATCT TCTTTTAAATTACATCATTAAATAACTCTG |
| Sequence-based reagent | Fzd10-PL-1 | This paper | SCRINSHOT padlock probe | /5Phos/CTCAAGGTAGACTTCATGTATCCTCTATGAT TACTGACTGCGTCTATTTAGTGGAGCCGCCCCTATC TTCTTTTACATTCAGCTGTTCTGAAC |
| Sequence-based reagent | Fzd10-PL-2 | This paper | SCRINSHOT padlock probe | /5Phos/TGGAGAGGAAGATGATAGGATCCTCTATGA TTACTGACTGCGTCTATTTAGTGGAGCCGCCCCTA TCTTCTTTCGAATAAACGCAGTAGCACA |
| Sequence-based reagent | Fzd10-PL-3 | This paper | SCRINSHOT padlock probe | /5Phos/GATTGTTCATCTTACACTTGTGTCCTCTAT GATTACTGACTGCGTCTATTTAGTGGAGCCGCCC CTATCTTCTTTCAGTCAGGTGTCTTGGTCT |
| Sequence-based reagent | Ctnnb1_Pl-1 | This paper | SCRINSHOT padlock probe | /5Phos/GTGTCAACATCTTCTTCCTCTCCTCTATG ATTACTGACTGCGTCTATTTAGTGGAGCCATGCC TATCTTCTTTCATTCATAAAGGACTTGGGAG |
| Sequence-based reagent | Ctnnb1_Pl-2 | This paper | SCRINSHOT padlock probe | /5Phos/GTCGCTGCATCTGAAAGGTTTCCTCTATG ATTACTGACTGCGTCTATTTAGTGGAGCCATGCC TATCTTCTTTCTTCCATCCCTTCCTGCTTA |
| Sequence-based reagent | Ctnnb1_Pl-3 | This paper | SCRINSHOT padlock probe | /5Phos/CAGGCCAGCTGATTGCTATCCTCTATGATT ACTGACTGCGTCTATTTAGTGGAGCCATGCCTATC TTCTTTGATTTACAGGTCAGTATCAAAC |

*Appendix 1 Continued on next page*

*Appendix 1 Continued*

| Reagent type (species) or resource | Designation | Source or reference | Identifiers | Additional information |
|---|---|---|---|---|
| Sequence-based reagent | Bmp4_Pl-1 | This paper | SCRINSHOT padlock probe | /5Phos/AGAATTAACAGAGTTGACTAGTCCTCTAT GATTACTGACTGCGTCTATTTAGTGGAGCCGCATC TATCTTCTTTAACAGGCCTTAGGGATACT |
| Sequence-based reagent | Bmp4_Pl-2 | This paper | SCRINSHOT padlock probe | /5Phos/GTGATGGAAACTCCTCACAGTCCTCTATGA TTACTGACTGCGTCTATTTAGTGGAGCCGCATCTAT CTTCTTTGATGTTCTCCAGATGTTCTTC |
| Sequence-based reagent | Bmp4_Pl-3 | This paper | SCRINSHOT padlock probe | /5Phos/GTGGTTGAATGGGAACGTGTCCTCTATGATT ACTGACTGCGTCTATTTAGTGGAGCCGCATCTATCTT CTTTCAGTTTGTGTGGTATGTGTAG |
| Sequence-based reagent | Bmp7_Pl-1 | This paper | SCRINSHOT padlock probe | /5Phos/TTCTTCAGGATGACATTAGAGTCCTCTATG ATTACTGACTGCGTCTATTTAGTGGAGCCTACGCTA TCTTCTTTGACCACCATGTTTCTGTAC |
| Sequence-based reagent | Bmp7_Pl-2 | This paper | SCRINSHOT padlock probe | /5Phos/AGTCCTTATAGATCCTGAATTCTCCTCTATGA TTACTGACTGCGTCTATTTAGTGGAGCCTACGCTAT CTTCTTTCAAATCGCTCCCGGATGT |
| Sequence-based reagent | Bmp7_Pl-3 | This paper | SCRINSHOT padlock probe | /5Phos/TGTCGCAGAACCTTCTGTTATCCTCTATGA TTACTGACTGCGTCTATTTAGTGGAGCCTACGCTA TCTTCTTTCTCAGATCCAGAAACCAATC |
| Sequence-based reagent | Smad6_PL-1 | This paper | SCRINSHOT padlock probe | /5Phos/ACTAAGGCTTGTGGATAACTCTCCTCTATG ATTACTGACTGCGTCTATTTAGTGGAGCCGCCCCTA TCTTCTTTTTTCCCTGCAGGATCCGAA |
| Sequence-based reagent | Smad6_PL-2 | This paper | SCRINSHOT padlock probe | /5Phos/CAGTGTAAGACAATGTAGAATCTCCTCT ATGA TTACTGACTGCGTCTATTTAGTGGAGCCGCCCCTAT C TTCTTTAGTTGGTGGCCTCGGTTT |
| Sequence-based reagent | Smad6_PL-3 | This paper | SCRINSHOT padlock probe | /5Phos/AGTAGGATCTCCAGCCAACTCCTCTATGATT ACTGACTGCGTCTATTTAGTGGAGCCGCCCCTATCT TCTTTCATTGCTATCTGTGGTTGTTG |
| Sequence-based reagent | Smad7_PL-1 | This paper | SCRINSHOT padlock probe | /5Phos/GCCTGCAGTTGGTTTGAGAATCCTCTAT GATT ACTGACTGCGTCTATTTAGTGGAGCCGCCCCTATCT T CTTTGAAGGTACAGCATCTGGACA |
| Sequence-based reagent | Smad7_PL-2 | This paper | SCRINSHOT padlock probe | /5Phos/GTACCAGCTGACTCTTGTTGTCCTCTATGATT ACTGACTGCGTCTATTTAGTGGAGCCGCCCCTATCT T CTTTGATCTTGCTCCGCACTTTCT |
| Sequence-based reagent | Smad7_PL-3 | This paper | SCRINSHOT padlock probe | /5Phos/CCTCTTGACTTCCGAGGAATCCTCTATGATT ACTGACTGCGTCTATTTAGTGGAGCCGCCCCTATCT TCTTTGTAAGATTCACAGCAACACAG |
| Sequence-based reagent | Nog_PL-1 | This paper | SCRINSHOT padlock probe | /5Phos/AGGGTCTGGATGTTCGATGATCCTCTAT GATT ACTGACTGCGTCTATTTAGTGGAGCCGCCCCTATCT T CTTTCTTCTCCTTAGGGTCAAAGAT |
| Sequence-based reagent | Nog_PL-2 | This paper | SCRINSHOT padlock probe | /5Phos/GATATAAATAGCAGGTTAACATTATCCTCTAT GA TTACTGACTGCGTCTATTTAGTGGAGCCGCCCCTAT CTT CTTTCATGCGAAGGGTACTGG |
| Sequence-based reagent | Nog_PL-3 | This paper | SCRINSHOT padlock probe | /5Phos/TGCATTACAGGAACCAGAAATCCTCTAT GATTAC TGACTGCGTCTATTTAGTGGAGCCGCCCCTATCTTC TTT CATTCCTACACAGTTAAACAG |

*Appendix 1 Continued on next page*

*Appendix 1 Continued*

| Reagent type (species) or resource | Designation | Source or reference | Identifiers | Additional information |
|---|---|---|---|---|
| Sequence-based reagent | mKi67_PL_1 | This paper | SCRINSHOT padlock probe | /5Phos/GGACTTTCCTGGAAATTCTGTCCTCTATGATTACT GACTGCGTCTATTTAGTGGAGCCGCCCCTATCTTCTTTC TTGATGGTTCCTTTCCAAG |
| Sequence-based reagent | mKi67_PL_2 | This paper | SCRINSHOT padlock probe | /5Phos/ACTTTCCTTGCATCTTTGAGTCCTCTATGATTAC TGACTGCGTCTATTTAGTGGAGCCGCCCCTATCTTCTT TGAACCTTCACTTAATTCACC |
| Sequence-based reagent | mKi67_PL_3 | This paper | SCRINSHOT padlock probe | /5Phos/CTGAGGCTTTATCAAGATTTCTCCTCTATGATT ACTGACTGCGTCTATTTAGTGGAGCCGCCCCTATCTTC TTTTCAGAGACTCCTTTCTCTTC |
| Sequence-based reagent | Cdkn1c_PL_1 | This paper | SCRINSHOT padlock probe | /5Phos/CAAGTTTAAGAGCAATCTAATGTCCTCTATGAT TACTGACTGCGTCTATTTAGTGGAGCCGCCCCTATCT TCTTTTTTCGACTGTCTGGTCAC |
| Sequence-based reagent | Cdkn1c_PL_2 | This paper | SCRINSHOT padlock probe | /5Phos/CGCATGGATAAATACAAATATTAATCCTCTATG ATTACTGACTGCGTCTATTTAGTGGAGCCGCCCCTAT CTTCTTTGTGGCAGAGGGATCCA |
| Sequence-based reagent | Cdkn1c_PL_3 | This paper | SCRINSHOT padlock probe | /5Phos/GCCTCTAAACTAACTCATCTCTCCTCTATGATT ACTGACTGCGTCTATTTAGTGGAGCCGCCCCTATCTT CTTTCAAGTTCTCTCTGGCCGTTA |
| Sequence-based reagent | Id3-1 | This paper | SCRINSHOT padlock probe | /5Phos/AGAGTAGAGATAGAGAGGGAGTCCTCTATGA TTACTGACTGCGTCTATTTAGTGGAGCCGCCCCTATC TTCTTTCTCAGCGCCTTCATGTTGG |
| Sequence-based reagent | Id3-2 | This paper | SCRINSHOT padlock probe | /5Phos/GGGCTGGGTTAAGATCGAATCCTCTATGATTA CTGACTGCGTCTATTTAGTGGAGCCGCCCCTATCTT CTTTGAGTTCAGGGTAAGTGAAGA |
| Sequence-based reagent | Id3-3 | This paper | SCRINSHOT padlock probe | /5Phos/TGCTGCCTTGGCAGAGTGTCCTCTATGATTA CTGACTGCGTCTATTTAGTGGAGCCGCCCCTATCTT CTTTGAAACCAGAAGAACAGCTCTTA |
| Sequence-based reagent | Spc24-1 | This paper | SCRINSHOT padlock probe | /5Phos/GCTGCTCTTCTATAGAATACTCCTCTATGAT TACTGACTGCGTCTATTTAGTGGAGCCGCCCCTAT CTTCTTTCTCAACAATTAAGAGCACTG |
| Sequence-based reagent | Spc24-2 | This paper | SCRINSHOT padlock probe | /5Phos/AAATCAATGAACGCACTCACTCCTCTATG ATTACTGACTGCGTCTATTTAGTGGAGCCGCCCC TATCTTCTTTCTGTCTCAAAGCAAGTAGAT |
| Sequence-based reagent | Spc24-3 | This paper | SCRINSHOT padlock probe | /5Phos/CACAGCTAACAGACACACATTCCTCTATG ATTACTGACTGCGTCTATTTAGTGGAGCCGCCCC TATCTTCTTTCTCTAACCTCTCCTGGAATA |
| Sequence-based reagent | Nrarp-1 | This paper | SCRINSHOT padlock probe | /5Phos/GGTACCATGCTCTCAGAATCCTCTATGATT ACTGACTGCGTCTATTTAGTGGAGCCGCCCCTAT CTTCTTTTATAATAGAAGTCACTAGGAAG |

*Appendix 1 Continued on next page*

*Appendix 1 Continued*

| Reagent type (species) or resource | Designation | Source or reference | Identifiers | Additional information |
|---|---|---|---|---|
| Sequence-based reagent | Nrarp-2 | This paper | SCRINSHOT padlock probe | /5Phos/ACTTCATACAGATAGAAGAGACTCCTCTATGATTACTGACTGCGTCTATTTAGTGGAGCCGCCCCTATCTTCTTTCTTAGGAGTGTGCCACAG |
| Sequence-based reagent | Nrarp-3 | This paper | SCRINSHOT padlock probe | /5Phos/TAGTATGAGAGATTCACAGTCTCCTCTATGATTACTGACTGCGTCTATTTAGTGGAGCCGCCCCTATCTTCTTTCCTTCCACTCATTCAGCAA |
| Sequence-based reagent | Rassf4-1 | This paper | SCRINSHOT padlock probe | /5Phos/ACTGAGGTCTTGTGGTTATAAATCCTCTATGATTACTGACTGCGTCTATTTAGTGGAGCCGCCCCTATCTTCTTTCCATAGGCAGGAGTGAAC |
| Sequence-based reagent | Rassf4-2 | This paper | SCRINSHOT padlock probe | /5Phos/CATCAAGAAGATCTTGACAATTTTCCTCTATGATTACTGACTGCGTCTATTTAGTGGAGCCGCCCCTATCTTCTTTCTCGCTCAAGTCAGCTTC |
| Sequence-based reagent | Rassf4-3 | This paper | SCRINSHOT padlock probe | CTCGGTGTCTTAGCTGAAAGTCCTCTATGATTACTGACTGCGTCTATTTAGTGGAGCCGCCCCTATCTTCTTTGATCAGAGTCCCTTCTTCCT |
| Sequence-based reagent | Nuak2-1 | This paper | SCRINSHOT padlock probe | /5Phos/TTGCTGCTATTCTCAAACACTCCTCTATGATTACTGACTGCGTCTATTTAGTGGAGCCGCCCCTATCTTCTTTACTCCATGACAATCACAATC |
| Sequence-based reagent | Nuak2-2 | This paper | SCRINSHOT padlock probe | /5Phos/GGACTTCTTAAGAGAATGTTGTCCTCTATGATTACTGACTGCGTCTATTTAGTGGAGCCGCCCCTATCTTCTTTCCATGTCATTCTCCTTTCG |
| Sequence-based reagent | Nuak2-3 | This paper | SCRINSHOT padlock probe | /5Phos/GTCGAGACTTCTTAAGGATGTCCTCTATGATTACTGACTGCGTCTATTTAGTGGAGCCGCCCCTATCTTCTTTGTAGTAACCAGATTCACGCT |
| Sequence-based reagent | Fam83d-1 | This paper | SCRINSHOT padlock probe | /5Phos/AATTGTCCGAACTGTCATCTCCTCTATGATTACTGACTGCGTCTATTTAGTGGAGCCGCCCCTATCTTCTTTTGCATAGTAGATATTTCCTGT |
| Sequence-based reagent | Fam83d-2 | This paper | SCRINSHOT padlock probe | /5Phos/GTAGAAGCCTCAGAGTCAGTCCTCTATGATTACTGACTGCGTCTATTTAGTGGAGCCGCCCCTATCTTCTTTGAAATAGTCTTCATCGCTGATG |
| Sequence-based reagent | Fam83d-3 | This paper | SCRINSHOT padlock probe | /5Phos/GTTGGTAGAGGCTCTGGTAATCCTCTATGATTACTGACTGCGTCTATTTAGTGGAGCCGCCCCTATCTCTTTATGTCCCTCACAGAAACCAG |
| Sequence-based reagent | Mybl2-1 | This paper | SCRINSHOT padlock probe | /5Phos/AAGTAGCAGCTATGGCAATCTCCTCTATGATTACTGACTGCGTCTATTTAGTGGAGCCGCCCCTATCTTCTTTATGTCCGAGTTCTTTAGCAG |
| Sequence-based reagent | Mybl2-2 | This paper | SCRINSHOT padlock probe | /5Phos/TTCTGATGATGGATACTTCTGTCCTCTATGATTACTGACTGCGTCTATTTAGTGGAGCCGCCCCTATCTCTTTACTTCTGATCTGGGAGTAC |

*Appendix 1 Continued on next page*

*Appendix 1 Continued*

| Reagent type (species) or resource | Designation | Source or reference | Identifiers | Additional information |
|---|---|---|---|---|
| Sequence-based reagent | Mybl2-3 | This paper | SCRINSHOT padlock probe | /5Phos/TTGACCAACTCAATGACCTTTTCCTCTATGATT ACTGACTGCGTCTATTTAGTGGAGCCGCCCCTATCT TC TTTCTGTTTGGTGCCATACTTC |
| Sequence-based reagent | Gas2-1 | This paper | SCRINSHOT padlock probe | /5Phos/TTCCTGAAGACTTTGATGAATCCTCTATGATT A CTGACTGCGTCTATTTAGTGGAGCCGCCCCTATCTT CT TTTAAGTTTCCAGTACTCTTCT |
| Sequence-based reagent | Gas2-2 | This paper | SCRINSHOT padlock probe | /5Phos/CTGTTATCTCTTTCCCTAATATCCTCTATGAT TA CTGACTGCGTCTATTTAGTGGAGCCGCCCCTATCTT C TTTTTCTCCATGAAGGTTTCTG |
| Sequence-based reagent | Gas2-3 | This paper | SCRINSHOT padlock probe | /5Phos/TCTTTGAATTTCTCCTGCACATCCTCTATGAT TACTGACTGCGTCTATTTAGTGGAGCCGCCCCTATC TTCTTTCTTGTTGGCATCCATACTC |
| Sequence-based reagent | Ada-1 | This paper | SCRINSHOT padlock probe | /5Phos/GAGAGCTTCATCCTCGATTCCTCTATGATTAC TGACTGCGTCTATTTAGTGGAGCCGCCCCTATCTTC TTTTTTCTTTCAGTAGTCTGTTGTA |
| Sequence-based reagent | Ada-2 | This paper | SCRINSHOT padlock probe | /5Phos/AAACTTGGCCAGGAAGCCTTCCTCTATGATT ACTGACTGCGTCTATTTAGTGGAGCCGCCCCTATCT TCTTTAATCACAGGCATGTAGTAGTC |
| Sequence-based reagent | Ada-3 | This paper | SCRINSHOT padlock probe | /5Phos/CAGCTCCAACACCTCAAGTCCTCTATGA TTAC TGACTGCGTCTATTTAGTGGAGCCGCCCCTATCTTC T TTTCTTCTGATTGTACTTCTTACA |
| Sequence-based reagent | Hspg2-1 | This paper | SCRINSHOT padlock probe | /5Phos/CTTCCTGTAGGCATCTTTGTCCTCTATGAT TACTGACTGCGTCTATTTAGTGGAGCCGCCCCTA TCTTCTTTGAAGGTGATCTTGATCTCAAA |
| Sequence-based reagent | Hspg2-2 | This paper | SCRINSHOT padlock probe | /5Phos/TGGATATCAGGGTCCAAGATCCTCTATGA TTACTGACTGCGTCTATTTAGTGGAGCCGCCCCTA TCTTCTTTATGATGTTATTGCCCGTAATC |
| Sequence-based reagent | Hspg2-3 | This paper | SCRINSHOT padlock probe | /5Phos/CTTGAGCCATGTACCTGATGTCCTCTATGA TTACTGACTGCGTCTATTTAGTGGAGCCGCCCCTA TCTTCTTTTCACCTGGAGAAGTCTCAGT |
| Sequence-based reagent | shc4-1 | This paper | SCRINSHOT padlock probe | /5Phos/TCATGGTCTCTGTCTTCAGGTCCTCTATGAT TACTGACTGCGTCTATTTAGTGGAGCCGCCCCTATC TTCTTTCCGGGATGGAGTTGTAATAG |
| Sequence-based reagent | shc4-2 | This paper | SCRINSHOT padlock probe | /5Phos/AGTCCTGTTTAGATTGGATTCTGTCCTCTATG ATTACTGACTGCGTCTATTTAGTGGAGCCGCCCCTA TCTTCTTTGGTGCAGGTGCAGGTCC |
| Sequence-based reagent | shc4-3 | This paper | SCRINSHOT padlock probe | /5Phos/GGTTGTCAAGGTTCATCAGTCCTCTATG ATTA CTGACTGCGTCTATTTAGTGGAGCCGCCCCTATCTT C TTTGGTGATTTGCAATAATCTGTT |
| Sequence-based reagent | Nuf2-1 | This paper | SCRINSHOT padlock probe | /5Phos/AATCACTTTCAATCTGGTCCTCCTCTATGATT ACTGACTGCGTCTATTTAGTGGAGCCGCCCCTATCT TCTTTCAGTTTCTTTAGTTCCGATG |
| Sequence-based reagent | Nuf2-2 | This paper | SCRINSHOT padlock probe | /5Phos/GGCTTTCATGTAAATCATGTTCCTCTATGATT ACTGACTGCGTCTATTTAGTGGAGCCGCCCCTATCT TCTTTACTCCATACACTAACTGTAA |

*Appendix 1 Continued*

| Reagent type (species) or resource | Designation | Source or reference | Identifiers | Additional information |
|---|---|---|---|---|
| Sequence-based reagent | Nuf2-3 | This paper | SCRINSHOT padlock probe | /5Phos/GTGGTTACTTGCTCACAGTCCTCTATGATTA CTGACTGCGTCTATTTAGTGGAGCCGCCCCTATCTT CTTTCTTGTGGATTTCTTGATTAACA |
| Sequence-based reagent | Ndc80-1 | This paper | SCRINSHOT padlock probe | /5Phos/ATACTCTCTGTCGGCTTTATCCTCTAT GATTACTGACTGCGTCTATTTAGTGGAGCCGC CCCTATCTTCTTTATCTTCAGACATGAATTCTTC |
| Sequence-based reagent | Ndc80-2 | This paper | SCRINSHOT padlock probe | /5Phos/GGATCCATGTCCGCTAGTCCTCTATGAT TACTGACTGCGTCTATTTAGTGGAGCCGCCCC TATCTTCTTTAATATACCAAGTTGACTATTCCT |
| Sequence-based reagent | Ndc80-3 | This paper | SCRINSHOT padlock probe | /5Phos/CTGTTCATTTAGTGCTTTGTTTTCCTCT ATGATTACTGACTGCGTCTATTTAGTGGAGCCG CCCCTATCTTCTTTCTCCTCCAGTCTTGCAAT |
| Sequence-based reagent | Spc25-1 | This paper | SCRINSHOT padlock probe | /5Phos/CCCATGTTTAGGTAGACTGTCCTCTAT GATTACTGACTGCGTCTATTTAGTGGAGCCGC CCCTATCTTCTTTAGATACCTCTATCATCTAGGT |
| Sequence-based reagent | Spc25-2 | This paper | SCRINSHOT padlock probe | /5Phos/ACAGATCCGCTGATTTCTGTCCTCTAT GATTACTGACTGCGTCTATTTAGTGGAGCCGC CCCTATCTTCTTTCTAGTCCAAGGTAATCTCTAT |
| Sequence-based reagent | Spc25-3 | This paper | SCRINSHOT padlock probe | /5Phos/AAATGGTTTCCCTCTTCCTATCCTCTAT GATTACTGACTGCGTCTATTTAGTGGAGCCGC CCCTATCTTCTTTATTAGCTTTGTTAGCAGTGG |
| Sequence-based reagent | Prdm16_Det_4 | This paper | SCRINSHOT detection probe | TCGCUACCCAAGUCTTCAGA/3Cy3Sp/ |
| Sequence-based reagent | Ttr_Det-1 | This paper | SCRINSHOT detection probe | TACTCUGTACACUCCTTCUACAAACTTC-FITC |
| Sequence-based reagent | Foxj1_Det_1 | This paper | SCRINSHOT detection probe | AGAAGTUGTCCGTGAUCCAC/3Cy5Sp/ |
| Sequence-based reagent | Foxj1_Det_2 | This paper | SCRINSHOT detection probe | AGGAAGGAUGUGGCCAAG/3Cy5Sp/ |
| Sequence-based reagent | Foxj1_Det_3 | This paper | SCRINSHOT detection probe | CGAGGCACUTGAUGAAGCA/3Cy5Sp/ |
| Sequence-based reagent | Sox2_Det-2 | This paper | SCRINSHOT detection probe | ATATAUACATGGATUCTCGGCAGC-cy3 |
| Sequence-based reagent | Sox2_Det-3 | This paper | SCRINSHOT detection probe | CGGAGTCUAGCTCTAAAUATTACTGG-cy3 |
| Sequence-based reagent | Ngn2_Det-1 | This paper | SCRINSHOT detection probe | GGGAAAGUTTGGTTUGACAGGT-Cy3 |
| Sequence-based reagent | Ngn2_Det-2 | This paper | SCRINSHOT detection probe | GCCTAAAUTTCCACGCUTGC-Cy3 |
| Sequence-based reagent | Ngn2_Det-3 | This paper | SCRINSHOT detection probe | ATCCTCACAUCTTCAAUAAACCAGAG-Cy3 |
| Sequence-based reagent | Hes5_Det-1 | This paper | SCRINSHOT detection probe | CCCACAUGACCAAGAGUTCAAAT-Cy3 |
| Sequence-based reagent | Hes5_Det-2 | This paper | SCRINSHOT detection probe | CAACTCUGCACACACAUTCTCT-Cy3 |
| Sequence-based reagent | Hes5_Det-3 | This paper | SCRINSHOT detection probe | CAAAGGAAUCCTTUAAGGAUCATCGT-Cy3 |
| Sequence-based reagent | Zbtb20_Det-1 | This paper | SCRINSHOT detection probe | GTCATAGUCATCTTCCAUTTCCTGC/3Cy5Sp/ |

*Appendix 1 Continued on next page*

*Appendix 1 Continued*

| Reagent type (species) or resource | Designation | Source or reference | Identifiers | Additional information |
|---|---|---|---|---|
| Sequence-based reagent | Zbtb20_Det-2 | This paper | SCRINSHOT detection probe | GAGAAAGGUACTACTTAUCCGTCAG/3Cy5Sp/ |
| Sequence-based reagent | Zbtb20_Det-3 | This paper | SCRINSHOT detection probe | AGTTTGUCAGTGAAUGTGTCGA/3Cy5Sp/ |
| Sequence-based reagent | Wnt2b_Det-1 | This paper | SCRINSHOT detection probe | CAACUCAAAUCACTCUGAAGACAGA-Cy5 |
| Sequence-based reagent | Wnt2b_Det-2 | This paper | SCRINSHOT detection probe | CGAACACCGUAATGGAUGTTGT-Cy5 |
| Sequence-based reagent | Wnt2b_Det-3 | This paper | SCRINSHOT detection probe | ATCGUGGAACGUGCAGUAGT-Cy5 |
| Sequence-based reagent | Wnt3a_Det-1 | This paper | SCRINSHOT detection probe | TTCCUAGAGAACCCUCTCTACTC-Cy5 |
| Sequence-based reagent | Wnt3a_Det-2 | This paper | SCRINSHOT detection probe | GTCCTTAGGUATGAGACCCUCA-Cy5 |
| Sequence-based reagent | Wnt3a_Det-3 | This paper | SCRINSHOT detection probe | AAGCACCCUCTCATGTAUCTAGATA-Cy5 |
| Sequence-based reagent | Foxj1_Det_2 | This paper | SCRINSHOT detection probe | TTGGGCAAUAAGTUAAGAGCCA/3Cy3Sp/ |
| Sequence-based reagent | Foxj1_Det_3 | This paper | SCRINSHOT detection probe | CAGGGTTAUTCATACCUAGAGACCA/3Cy3Sp/ |
| Sequence-based reagent | Sox2_Det-2 | This paper | SCRINSHOT detection probe | CGGCCUGCTTCATUGTTG/3Cy3Sp/ |
| Sequence-based reagent | Sox2_Det-3 | This paper | SCRINSHOT detection probe | AGAGAAGUAAAGGGATUTGTCTCC/36-FAM/ |
| Sequence-based reagent | Ngn2_Det-1 | This paper | SCRINSHOT detection probe | CCCGCCUCATTGUTGTGA/36-FAM/ |
| Sequence-based reagent | Ngn2_Det-2 | This paper | SCRINSHOT detection probe | CTTGCTUCTCCCGGUCAC/36-FAM/ |
| Sequence-based reagent | Ngn2_Det-3 | This paper | SCRINSHOT detection probe | AGGTAAATGUCAGAAGUCAAGAACA/36-FAM/ |
| Sequence-based reagent | Hes5_Det-1 | This paper | SCRINSHOT detection probe | CGGAACCUACGTGAUAAGGATT/36-FAM/ |
| Sequence-based reagent | Hes5_Det-2 | This paper | SCRINSHOT detection probe | TCAGCAUCCTCCTGUATGGA/36-FAM/ |
| Sequence-based reagent | Hes5_Det-3 | This paper | SCRINSHOT detection probe | TCGCUACCCAAGUCTTCAGA/3Cy3Sp/ |
| Sequence-based reagent | Zbtb20_Det-1 | This paper | SCRINSHOT detection probe | TACTCUGTACACUCCTTCUACAAACTTC-FITC |
| Sequence-based reagent | Zbtb20_Det-2 | This paper | SCRINSHOT detection probe | AGAAGTUGTCCGTGAUCCAC/3Cy5Sp/ |
| Sequence-based reagent | Zbtb20_Det-3 | This paper | SCRINSHOT detection probe | AGGAAGGAUGUGGCCAAG/3Cy5Sp/ |
| Sequence-based reagent | Wnt2b_Det-1 | This paper | SCRINSHOT detection probe | CGAGGCACUTGAUGAAGCA/3Cy5Sp/ |
| Sequence-based reagent | Wnt2b_Det-2 | This paper | SCRINSHOT detection probe | ATATAUACATGGATUCTCGGCAGC-cy3 |
| Sequence-based reagent | Wnt2b_Det-3 | This paper | SCRINSHOT detection probe | CGGAGTCUAGCTCTAAAUATTACTGG-cy3 |

*Appendix 1 Continued on next page*

*Appendix 1 Continued*

| Reagent type (species) or resource | Designation | Source or reference | Identifiers | Additional information |
|---|---|---|---|---|
| Sequence-based reagent | Wnt3a_Det-1 | This paper | SCRINSHOT detection probe | GGGAAAGUTTGGTTUGACAGGT-Cy3 |
| Sequence-based reagent | Wnt3a_Det-2 | This paper | SCRINSHOT detection probe | GCCTAAAUTTCCACGCUTGC-Cy3 |
| Sequence-based reagent | Wnt3a_Det-3 | This paper | SCRINSHOT detection probe | ATCCTCACAUCTTCAAUAAACCAGAG-Cy3 |
| Sequence-based reagent | Wnt5a_Det-1 | This paper | SCRINSHOT detection probe | CCCACAUGACCAAGAGUTCAAAT-Cy3 |
| Sequence-based reagent | Wnt5a_Det-2 | This paper | SCRINSHOT detection probe | CAACTCUGCACACACAUTCTCT-Cy3 |
| Sequence-based reagent | Wnt5a_Det-3 | This paper | SCRINSHOT detection probe | CAAAGGAAUCCTTUAAGGAUCATCGT-Cy3 |
| Sequence-based reagent | Wnt7b_Det-1 | This paper | SCRINSHOT detection probe | GTCATAGUCATCTTCCAUTTCCTGC/3Cy5Sp/ |
| Sequence-based reagent | Wnt7b_Det-2 | This paper | SCRINSHOT detection probe | GAGAAAGGUACTACTTAUCCGTCAG/3Cy5Sp/ |
| Sequence-based reagent | Wnt7b_Det-3 | This paper | SCRINSHOT detection probe | AGTTTGUCAGTGAAUGTGTCGA/3Cy5Sp/ |
| Sequence-based reagent | Axin2_Det-1 | This paper | SCRINSHOT detection probe | CAACUCAAAUCACTCUGAAGACAGA-Cy5 |
| Sequence-based reagent | Axin2_Det-2 | This paper | SCRINSHOT detection probe | CGAACACCGUAATGGAUGTTGT-Cy5 |
| Sequence-based reagent | Axin2_Det-3 | This paper | SCRINSHOT detection probe | ATCGUGGAACGUGCAGUAGT-Cy5 |
| Sequence-based reagent | Lef1_Det-1 | This paper | SCRINSHOT detection probe | TCGCTGUTCATATUGGGCAT-FITC |
| Sequence-based reagent | Lef1_Det-2 | This paper | SCRINSHOT detection probe | TCTCTCTTUCCGTGCUAGTTCA-FITC |
| Sequence-based reagent | Lef1_Det-3 | This paper | SCRINSHOT detection probe | GCTATCTCTUTAGCTTCAUCAGACAC-FITC |
| Sequence-based reagent | Tcf3_Det_1 | This paper | SCRINSHOT detection probe | GAAGUCCAGGAGGUCACTC/36-FAM/ |
| Sequence-based reagent | Tcf3_Det_2 | This paper | SCRINSHOT detection probe | GCTTCGAUGGAGACCUGTC/36-FAM/ |
| Sequence-based reagent | Tcf3_Det_3 | This paper | SCRINSHOT detection probe | GATTTCCUCATCCTCTUTCTCCTC/36-FAM/ |
| Sequence-based reagent | Fzd9-Det-1 | This paper | SCRINSHOT detection probe | GTAGACATUGTAGCACAUAGAAAGGA/36-FAM/ |
| Sequence-based reagent | Fzd9-Det-2 | This paper | SCRINSHOT detection probe | TGAGGCGTUCATAAACGUAGC/36-FAM/ |
| Sequence-based reagent | Fzd9-Det-3 | This paper | SCRINSHOT detection probe | TAAATAACUCTGCACUGGACCC/36-FAM/ |
| Sequence-based reagent | Fzd10-Det-1 | This paper | SCRINSHOT detection probe | GCTGTUCTGAACCUCAAGGTA/3Cy5Sp/ |
| Sequence-based reagent | Fzd10-Det-2 | | | CGCAGUAGCACAUGGAGAG/3Cy5Sp/ |
| Sequence-based reagent | Fzd10-Det-3 | This paper | SCRINSHOT detection probe | TGTCTTGGUCTGATTGTUCATCTT/3Cy5Sp/ |

*Appendix 1 Continued*

| Reagent type (species) or resource | Designation | Source or reference | Identifiers | Additional information |
|---|---|---|---|---|
| Sequence-based reagent | Ctnnb1_Det-1 | This paper | SCRINSHOT detection probe | GACUTGGGAGGUGTCAACAT-Cy5 |
| Sequence-based reagent | Ctnnb1_Det-2 | This paper | SCRINSHOT detection probe | CCTGCUTAGTCGCUGCA-Cy5 |
| Sequence-based reagent | Ctnnb1_Det-3 | This paper | SCRINSHOT detection probe | TTACAGGUCAGTAUCAAACCAGG-Cy5 |
| Sequence-based reagent | Bmp4_Det-1 | This paper | SCRINSHOT detection probe | GCCTTUGGGATACUAGAATUAACAGAG-FITC |
| Sequence-based reagent | Bmp4_Det-2 | This paper | SCRINSHOT detection probe | CAGAUGTTCTUCGTGAUGGAAACT-FITC |
| Sequence-based reagent | Bmp4_Det-3 | This paper | SCRINSHOT detection probe | GTGGUATGTGUAGGTGGUTGAATG-FITC |
| Sequence-based reagent | Bmp7_Det-1 | This paper | SCRINSHOT detection probe | CATGTTTCUGTACTTCTUCAGGATGA-Cy5 |
| Sequence-based reagent | Bmp7_Det-2 | This paper | SCRINSHOT detection probe | TCCCGGAUGTAGTCCUTATAGATC-Cy5 |
| Sequence-based reagent | Bmp7_Det-3 | This paper | SCRINSHOT detection probe | GAAACCAAUCTGUCGCAGAAC-Cy5 |
| Sequence-based reagent | Smad6_Det-1 | This paper | SCRINSHOT detection probe | CAGGAUCCGAAACUAAGGCTT/36-FAM/ |
| Sequence-based reagent | Smad6_Det-2 | This paper | SCRINSHOT detection probe | CCTCGGUTTCAGTGUAAGACAA/36-FAM/ |
| Sequence-based reagent | Smad6_Det-3 | This paper | SCRINSHOT detection probe | CTGTGGUTGTTGAGUAGGATCT/36-FAM/ |
| Sequence-based reagent | Smad7_Det-1 | This paper | SCRINSHOT detection probe | CUGGACAGCCUGCAGT/36-FAM/ |
| Sequence-based reagent | Smad7_Det-2 | This paper | SCRINSHOT detection probe | CGCACUTTCTGUACCAGCT/36-FAM/ |
| Sequence-based reagent | Smad7_Det-3 | This paper | SCRINSHOT detection probe | ACACAGCCUCTTGACUTCC/36-FAM/ |
| Sequence-based reagent | Nog_Det-1 | This paper | SCRINSHOT detection probe | GTCAAAGAUAGGGTCUGGATGTTC/3Cy5Sp/ |
| Sequence-based reagent | Nog_Det-2 | This paper | SCRINSHOT detection probe | GCGAAGGGUACTGGGATAUAAATAG/3Cy5Sp/ |
| Sequence-based reagent | Nog_Det-3 | This paper | SCRINSHOT detection probe | CTACACAGUTAAACAGUGCATTACAG/3Cy5Sp/ |
| Sequence-based reagent | mKi67_Det_1 | This paper | SCRINSHOT detection probe | CCTTUCCAAGGGACUTTCCT/3Cy3Sp/ |
| Sequence-based reagent | mKi67_Det_2 | This paper | SCRINSHOT detection probe | ACTTAATUCACCACTTUCCTTGCA/3Cy3Sp/ |
| Sequence-based reagent | mKi67_Det_3 | This paper | SCRINSHOT detection probe | CTTTCTCTUCCTGAGGCUTTATCA/3Cy3Sp/ |
| Sequence-based reagent | Cdkn1c_Det-1 | This paper | SCRINSHOT detection probe | TCTGGUCACCAAGTUTAAGAGC/36-FAM/ |
| Sequence-based reagent | Cdkn1c_Det-2 | This paper | SCRINSHOT detection probe | GGGAUCCACGCAUGGATAA/36-FAM/ |
| Sequence-based reagent | Cdkn1c_Det-3 | This paper | SCRINSHOT detection probe | TGGCCGUTAGCCTCUAAAC/36-FAM/ |

*Appendix 1 Continued on next page*

*Appendix 1 Continued*

| Reagent type (species) or resource | Designation | Source or reference | Identifiers | Additional information |
|---|---|---|---|---|
| Sequence-based reagent | Id3-Det-1 | This paper | SCRINSHOT detection probe | CCTTCATGUTGGAGAGUAGAGATAG/3Cy3Sp/ |
| Sequence-based reagent | Id3-Det-2 | This paper | SCRINSHOT detection probe | TAAGUGAAGAGGGCUGGGT/3Cy3Sp/ |
| Sequence-based reagent | Id3-Det-3 | This paper | SCRINSHOT detection probe | GAACAGCUCTTATGCUGCCT/3Cy3Sp/ |
| Sequence-based reagent | Spc24-Det-1 | This paper | SCRINSHOT detection probe | GAGCACUGGCTGCUCTT/36-FAM/ |
| Sequence-based reagent | Spc24-Det-2 | This paper | SCRINSHOT detection probe | AAAGCAAGUAGATAAAUCAATGAACGC/36-FAM/ |
| Sequence-based reagent | Spc24-Det-3 | This paper | SCRINSHOT detection probe | CCTGGAAUACACAGCUAACAGAC/36-FAM/ |
| Sequence-based reagent | Nrarp-Det-1 | This paper | SCRINSHOT detection probe | AGTCACUAGGAAGGGUACCATG/3Cy5Sp/ |
| Sequence-based reagent | Nrarp-Det-2 | This paper | SCRINSHOT detection probe | GUGCCACAGACUTCAUACAGATAG/3Cy5Sp/ |
| Sequence-based reagent | Nrarp-Det-3 | This paper | SCRINSHOT detection probe | ACTCATUCAGCAATAGUATGAGAGAUTC/3Cy5Sp/ |
| Sequence-based reagent | Rassf4-Det-1 | This paper | SCRINSHOT detection probe | AGGAGUGAACACUGAGGTCT/3Cy3Sp/ |
| Sequence-based reagent | Rassf4-Det-2 | This paper | SCRINSHOT detection probe | TCAGCUTCCAUCAAGAAGAUCTTG/3Cy3Sp/ |
| Sequence-based reagent | Rassf4-Det-3 | This paper | SCRINSHOT detection probe | CCTTCUTCCTCUCGGTGTC/3Cy3Sp/ |
| Sequence-based reagent | Nuak2-Det-1 | This paper | SCRINSHOT detection probe | GACAAUCACAAUCTTGCTGCUATTC/36-FAM/ |
| Sequence-based reagent | Nuak2-Det-2 | This paper | SCRINSHOT detection probe | TCTCCTTUCGGGACTUCTTAAGA/36-FAM/ |
| Sequence-based reagent | Nuak2-Det-3 | This paper | SCRINSHOT detection probe | TCACGCTGUCGAGACUTCT/36-FAM/ |
| Sequence-based reagent | Fam83d-Det-1 | This paper | SCRINSHOT detection probe | GTAGATATTUCCTGTAATUGTCCGAACT/36-FAM/ |
| Sequence-based reagent | Fam83d-Det-2 | This paper | SCRINSHOT detection probe | TTCATCGCUGATGGUAGAAGC/36-FAM/ |
| Sequence-based reagent | Fam83d-Det-3 | This paper | SCRINSHOT detection probe | ACCAGGUTGGUAGAGGCT/36-FAM/ |
| Sequence-based reagent | Mybl2-Det-1 | This paper | SCRINSHOT detection probe | GTTCTTUAGCAGAAGUAGCAGCT/3Cy5Sp/ |
| Sequence-based reagent | Mybl2-Det-2 | This paper | SCRINSHOT detection probe | CTGGGAGUACTTCTGAUGATGG/3Cy5Sp/ |
| Sequence-based reagent | Mybl2-Det-3 | This paper | SCRINSHOT detection probe | GCCAUACTTCTUGACCAACUCAAT/3Cy5Sp/ |
| Sequence-based reagent | Gas2-Det-1 | This paper | SCRINSHOT detection probe | CCAGTACUCTTCTTTCCUGAAGAC/36-FAM/ |
| Sequence-based reagent | Gas2-Det-2 | This paper | SCRINSHOT detection probe | TGAAGGTTUCTGCTGTTAUCTCTTTC/36-FAM/ |
| Sequence-based reagent | Gas2-Det-3 | This paper | SCRINSHOT detection probe | CATCCATACUCTCTTTGAAUTTCTCCT/36-FAM/ |

*Appendix 1 Continued on next page*

*Appendix 1 Continued*

| Reagent type (species) or resource | Designation | Source or reference | Identifiers | Additional information |
|---|---|---|---|---|
| Sequence-based reagent | Ada-Det-1 | This paper | SCRINSHOT detection probe | AGTAGUCTGTTGUAGAGAGCUTCAT/36-FAM/ |
| Sequence-based reagent | Ada-Det-2 | This paper | SCRINSHOT detection probe | CATGTAGUAGTCAAACUTGGCCA/36-FAM/ |
| Sequence-based reagent | Ada-Det-3 | This paper | SCRINSHOT detection probe | GTACTTCUTACACAGCUCCAACA/36-FAM/ |
| Sequence-based reagent | Hspg2-Det-1 | This paper | SCRINSHOT detection probe | CTTGATCUCAAACTTCCUGTAGGC/3Cy3Sp/ |
| Sequence-based reagent | Hspg2-Det-2 | This paper | SCRINSHOT detection probe | GCCCGTAAUCTGGAUATCAGG/3Cy3Sp/ |
| Sequence-based reagent | Hspg2-Det-3 | This paper | SCRINSHOT detection probe | AGAAGTCUCAGTCTUGAGCCA/3Cy3Sp/ |
| Sequence-based reagent | Shc4-Det-1 | This paper | SCRINSHOT detection probe | GGAGTTGUAATAGTCAUGGTCTCTG/3Cy3Sp/ |
| Sequence-based reagent | Shc4-Det-2 | This paper | SCRINSHOT detection probe | GCAGGUCCAGTCCUGTTTAG/3Cy3Sp/ |
| Sequence-based reagent | Shc4-Det-3 | This paper | SCRINSHOT detection probe | GCAATAAUCTGTTGGTUGTCAAGG/3Cy3Sp/ |
| Sequence-based reagent | Nuf2-Det-1 | This paper | SCRINSHOT detection probe | AGTTCCGAUGAATCACTUTCAATCT/3Cy5Sp/ |
| Sequence-based reagent | Nuf2-Det-2 | This paper | SCRINSHOT detection probe | TACACTAACUGTAAGGCTUTCATGTAA/3Cy5Sp/ |
| Sequence-based reagent | Nuf2-Det-3 | This paper | SCRINSHOT detection probe | ATTTCTTGAUTAACAGTGGUTACTTGC/3Cy5Sp/ |
| Sequence-based reagent | Ndc80-1 | This paper | SCRINSHOT detection probe | GACATGAATUCTTCATACUCTCTGTCG/3Cy3Sp/ |
| Sequence-based reagent | Ndc80-2 | This paper | SCRINSHOT detection probe | CAAGTUGACTATTCCUGGATCCA/3Cy3Sp/ |
| Sequence-based reagent | Ndc80-3 | This paper | SCRINSHOT detection probe | GTCTTGCAAUCTGTTCATUTAGTGC/3Cy3Sp/ |
| Sequence-based reagent | Spc25-Det-1 | This paper | SCRINSHOT detection probe | TCTATCATCUAGGTCCCAUGTTTAGG/3Cy5Sp/ |
| Sequence-based reagent | Spc25-Det-2 | This paper | SCRINSHOT detection probe | AGGTAATCUCTATACAGAUCCGCTG/3Cy5Sp/ |
| Sequence-based reagent | Spc25-Det-3 | This paper | SCRINSHOT detection probe | GTTAGCAGUGGAAATGGUTTCC/3Cy5Sp/ |

