## [Editor Report · eLife Assessment]

This **valuable** work presents how PRDM16 plays a critical role during colloid plexus development, through regulating BMP signaling. **Solid** evidence supports the context-dependent gene regulatory mechanisms both in vivo and in vitro. The work will be of broad interest to researchers working on growth factor signaling mechanisms and vertebrate development.

---

## [Referee Report · Reviewer #1 (Public review)]

Summary:

This manuscript describes the role of PRDM16 in modulating BMP response during choroid plexus (ChP) development. The authors combine PRDM16 knockout mice and cultured PRDM16 KO primary neural stem cells (NSCs) to determine the interactions between BMP signaling and PRDM16 in ChP differentiation.

They show PRDM16 KO affects ChP development in vivo and BMP4 response in vitro. They determine genes regulated by BMP and PRDM16 by ChIP-seq or CUT&TAG for PRDM16, pSMAD1/5/8, and SMAD4. They then measure gene activity in primary NSCs through H3K4me3 and find more genes are corepressed than coactivated by BMP signaling and PRDM16 and focus on the 31 genes found to be co-repressed by BMP and PRDM16. Wnt7b is in this set and the authors then provide evidence that PRDM16 and BMP signaling together repress Wnt activity in the developing choroid plexus.

Strengths:

Understanding context-dependent response to cell signals during development is an important problem. The authors use a powerful combination of in vivo and in vitro systems to dissect how PRDM16 may modulate BMP response in early brain development.

Main weakness of the experimental setup:

(1) Because the authors state that primary NSCs cultured in vitro lose endogenous Prdm16 expression, they drive expression by a constitutive promoter. However, this means the expression levels is very different from endogenous levels (as explicitly shown in Supp. Fig. 2B) and the effect of many transcription factors is strongly dose-dependent, likely creating differences between the PRDM16-dependent transcriptional response in the in vitro system and in vivo. Although the authors combine in vitro and in vivo evidence on the role of PRDM16 as a co-factor for MBP signaling and verified that BMP induces quiescence in their NSC model in a PRDM16-dependent manner, this experimental setup remains a weakness and likely affects the results of the various genomics experiments.

Other experimental weaknesses that make the evidence less convincing:

(1) It seems that the authors compare Prdm16_KO cells to Prdm16 WT cells overexpressing flag_Prdm16. Aside from the possible expression of endogenous Prdm16, other cell differences may have arisen between these cell lines. A properly controlled experiment would compare Prdm16_KO ctrl (possibly infected with a control vector without Prdm16) to Prdm16_KO_E (i.e. the Prdm16_KO cells with and without Prdm16 overexpression.) The authors acknowledged this problem in their rebuttal, stating that they were unable to overexpress PRDM16 in KO cells.

(2) The authors show in Fig.2E that Ttr is not upregulated by BMP4 in PRDM16_KO NSCs. This appears inconsistent with the presence of Ttr expression in the PRDM16_KO brain in Fig.1C. The authors explained in their rebuttal that the Ttr protein levels are not detectable in the NSCs with antibodies but the effect is still visible at the level of mRNA. The dramatic difference in protein expression is curious.

---

## [Referee Report · Reviewer #2 (Public review)]

The authors have revised their manuscript in response to reviewer feedback, incorporating several modifications to improve clarity and provide additional supporting information. To address concerns about confusing terminology, they have standardized the reference to PRDM16 overexpressing cells as Prdm16_OE, clarifying its expression from a constitutive promoter. They also revised the text to resolve seemingly contradictory statements about ChP development in the mutant. New bioinformatic analysis comparing PRDM16 binding in E12.5 ChP cells to co-repressed versus BMP-only-repressed genes has been performed and included in Supplementary Figure 5C, providing a statistical assessment of PRDM16's regulatory role on co-repressed genes. Several figures were updated, including adding an illustration of the Prdm16 cGT allele to Figure 1B, providing a zoomed-in inset for Figure 1E, and including individual channels for Wnt2b and marking boundaries in Figure 7A. Full-view images and examples of spot segmentation for SCRINSHOT analysis are now available in a new supplementary figure, and the presentation of RT-qPCR data in Supplementary Figure 2B was improved by using separate graphs for overexpression samples to avoid a broken Y-axis. Furthermore, the authors have added more references to introductory statements, annotated structures like the ChP, CH, and fourth ventricle in figures, and clarified that the beta-Gal signal was used as a marker for mutant ChP cells in Figure 1D. Finally, the manuscript now includes a discussion of the recently published, related study by Hurwitz et al. (2023) in the discussion section, highlighting similarities and differences. Overall, the authors have satisfactorily addressed the reviewers' comments.

---

## [Referee Report · Reviewer #3 (Public review)]

Summary:

Bone morphogenetic protein (BMP) signaling instructs multiple processes during development including cell proliferation and differentiation. The authors set out to understand the role of PRDM16 in these various functions of BMP signaling. They find that PRDM16 and BMP co-operate to repress stem cell proliferation by regulating the genomic distribution of BMP pathway transcription factors. They additionally show that PRDM16 impacts choroid plexus epithelial cell specification. The authors provide evidence for a regulatory circuit (constituting of BMP, PRDM16 and Wnt) that influences stem cell proliferation/differentiation.

Strengths:

I find the topics studied by the authors in this study of general interest to the field, the experiments well-controlled and the analysis in the paper sound. I have no major scientific concerns.

Weaknesses:

I have some minor recommendations which will help improve the paper (regarding the discussion).

Comments on revised version:

The authors have addressed my concerns in the revised version of the manuscript.

---

## [Author Response]

The following is the authors’ response to the original reviews

**Reviewer #1 (Public review):**
Summary:This manuscript describes the role of PRDM16 in modulating BMP response during choroid plexus (ChP) development. The authors combine PRDM16 knockout mice and cultured PRDM16 KO primary neural stem cells (NSCs) to determine the interactions between BMP signaling and PRDM16 in ChP differentiation.They show PRDM16 KO affects ChP development in vivo and BMP4 response in vitro. They determine genes regulated by BMP and PRDM16 by ChIP-seq or CUT&TAG for PRDM16, pSMAD1/5/8, and SMAD4. They then measure gene activity in primary NSCs through H3K4me3 and find more genes are co-repressed than co-activated by BMP signaling and PRDM16. They focus on the 31 genes found to be co-repressed by BMP and PRDM16. Wnt7b is in this set and the authors then provide evidence that PRDM16 and BMP signaling together repress Wnt activity in the developing choroid plexus.Strengths:Understanding context-dependent responses to cell signals during development is an important problem. The authors use a powerful combination of in vivo and in vitro systems to dissect how PRDM16 may modulate BMP response in early brain development.

We thank the reviewer for the thoughtful summary and positive feedback. We appreciate the recognition of our integrative in vivo and in vitro approach. We're glad the reviewer found our findings on context-dependent gene regulation and developmental signaling valuable.

Main weaknesses of the experimental setup:(1) Because the authors state that primary NSCs cultured in vitro lose endogenous Prdm16 expression, they drive expression by a constitutive promoter. However, this means the expression levels are very different from endogenous levels (as explicitly shown in Supplementary Figure 2B) and the effect of many transcription factors is strongly dose-dependent, likely creating differences between the PRDM16-dependent transcriptional response in the in vitro system and in vivo.

We acknowledge that our in vitro experiments may not ideally replicate the in vivo situation, a common limitation of such experiments, our primary aim was to explore the molecular relationship between PRDM16 and BMP signaling in gene regulation. Such molecular investigations are challenging to conduct using in vivo tissues. In vitro NSCs treated with BMP4 has been used a model to investigate NSC proliferation and quiescence, drawing on previous studies (e.g., Helena Mira, 2010; Marlen Knobloch, 2017). Crucially, to ensure the relevance of our in vitro findings to the in vivo context, we confirmed that cultured cells could indeed be induced into quiescence by BMP4, and this induction necessitated the presence of PRDM16. Furthermore, upon identifying target genes co-regulated by PRDM16 and SMADs, we validated PRDM16's regulatory role on a subset of these genes in the developing Choroid Plexus (ChP) (Fig. 7 and Suppl.Fig7-8). Only by combining evidence from both in vitro and in vivo experiments could we confidently conclude that PRDM16 serves as an essential co-factor for BMP signaling in restricting NSC proliferation.

(2) It seems that the authors compare Prdm16_KO cells to Prdm16 WT cells overexpressing flag_Prdm16. Aside from the possible expression of endogenous Prdm16, other cell differences may have arisen between these cell lines. A properly controlled experiment would compare Prdm16_KO ctrl (possibly infected with a control vector without Prdm16) to Prdm16_KO_E (i.e. the Prdm16_KO cells with and without Prdm16 overexpression.)

We agree that Prdm16 KO cells carrying the Prdm16-expressing vector would be a good comparison with those with KO_vector. However, despite more than 10 attempts with various optimization conditions, we were unable to establish a viable cell line after infecting Prdm16 KO cells with the Prdm16-expressing vector. The overall survival rate for primary NSCs after viral infection is low, and we observed that KO cells were particularly sensitive to infection treatment when the viral vector was large (the Prdm16 ORF is more than 3kb).

As an alternative oo assess vector effects, we instead included two other control cell lines, wt and KO cells infected with the 3xNLS_Flag-tag viral vector, and presented the results in supplementary Fig 2. When we compared the responses of the four lines — wt, KO, wt infected with the Flag vector, KO infected with the Flag vector — to the addition and removal of BMP4, we confirmed that the viral infection itself has no significant impacts on the responses of these cells to these treatments regarding changes in cell proliferation and Ttr induction.

Given that wt cells and the KO cells, with or without viral backbone infection behave quite similarly in terms of cell proliferation, we speculate that even if we were successful in obtaining a cell line with Prdm16-expressing vector in the KO cells, it may not exhibit substantial differences compared to wt cells infected with Prdm16-expressing vector.

Other experimental weaknesses that make the evidence less convincing:(1) The authors show in Figure 2E that Ttr is not upregulated by BMP4 in PRDM16_KO NSCs. Does this appear inconsistent with the presence of Ttr expression in the PRDM16_KO brain in Figure1C?

The reviwer’s point is that there was no significant increase in Ttr expression in Prdm16_KO cells after BMP4 treatment (Fig. 2E), but there remained residule Ttr mRNA signals in the Prdm16 mutant ChP (Fig. 1C). We think the difference lies in the measuable level of Ttr expression between that induced by BMP4 in NSC culture and that in the ChP. This is based on our immunostaining expreriment in which we tried to detect Ttr using a Ttr antibody. This antibody could not detect the Ttr protein in BMP4-treated Prdm16_expressing NSCs but clearly showed Ttr signal in the wt ChP. This means that although Ttr expression can be significantly increased by BMP4 in vitro to a level measurable by RT-qPCR, its absolute quantity even in the Prdm16_expressing condition is much lower compared to that in vivo. Our results in Fig 1C and Fig 2E, as well as Fig 7B, all consistently showed that Prdm16 depletion significantly reduced Ttr expression in in vitro and in vivo.

(2) Figure 3: The authors use H3K4me3 to measure gene activity. This is however, very indirect, with bulk RNA-seq providing the most direct readout and polymerase binding (ChIP-seq) another more direct readout. Transcription can be regulated without expected changes in histone methylation, see e.g. papers from Josh Brickman. They verify their H3K4me3 predictions with qPCR for a select number of genes, all related to the kinetochore, but it is not clear why these genes were picked, and one could worry whether these are representative.

H3K4me3 has widely been used as an indicator of active transcription and is a mark for cell identity genes. And it has been demonstrated that H3K4me3 has a direct function in regulating transciption at the step of RNApolII pausing release. As stated in the text, there are advantages and disadvantages of using H3K4me3 compared to using RNA-seq. RNA-seq profiles all gene products, which are affected by transcription and RNA stability and turnover. In contrast, H3K4me3 levels at gene promoter reflects transcriptional activity. In our case, we aimed to identify differential gene expression between proliferation and quiescence states. The transition between these two states is fast and dynamic. RNA-seq may not be able to identify functionally relevant genes but more likely produces false positive and negative results. Therefore, we chose H3K4me3 profiling.

We agree that transcription may change without histone methylation changes. This may cause an under-estimation of the number of changed genes between the conditions.

We validated 7 out of 31 genes (Wnt7b, Id3, Mybl2, Spc24, Spc25, Ndc80 and Nuf2). We chose these genes based on two critira: (1) their function is implicated in cell proliferation and cell-cycle regulation based on gene ontology analysis; (2) their gene products are detectable in the developing ChP based on the scRNA-seq data. Three of these genes (Wnt7b, Id3, Mybl2) are not related to the kinetochore. We now clarify this description in the revised text.

(3) Line 256: The overlap of 31 genes between 184 BMP-repressed genes and 240 PRDM16-repressed genes seems quite small.

This result indicates that in addition to co-repressing cell-cycle genes, BMP and PRDM16 have independent fucntions. For example, it was reported that BMP regulates neuronal and astrocyte differentiation (Katada, S. 2021), while our previous work demonstrated that Prdm16 controls temporal identity of NSCs (He, L. 2021).

(4) The Wnt7b H3K4me3 track in Fig. 3G is not discussed in the text but it shows H3K4me3 high in _KO and low in _E regardless of BMP4. This seems to contradict the heatmap of H3K4me3 in Figure 3E which shows H3K4me3 high in _E no BMP4 and low in _E BMP4 while omitting _KO no BMP4. Meanwhile CDKN1A, the other gene shown in 3G, is missing from 3E.

The track in Fig 3G shows the absolute signal of H3K4me3 after mapping the sequencing reads to the genome and normaliz them to library size. Compare the signal in Prdm16_E with BMP4 and that in Prdm16_E without BMP4, the one with BMP4 has a lower peak. The same trend can be seen for the pair of Prdm16_KO cells with or without BMP4. The heatmap in Fig. 3E shows the relative level of H3K4me3 in three conditions. The Prdm16_E cells with BMP4 has the lowest level, while the other two conditions (Prdm16_KO with BMP4 and Prdm16_E without BMP4) display higher levels. These two graphs show a consistent trend of H3K4me3 changes at the Wnt7b promoter across these conditions. Figure 3E only includes genes that are co-repressed by PRDM16 and BMP. CDKN1A’s H3K4me3 signals are consistent between the conditions, and thus it is not a PRDM16- or BMP-regulated gene. We use it as a negative control.

(5) The authors use PRDM16 CUT&TAG on dissected dorsal midline tissues to determine if their 31 identified PRDM16-BMP4 co-repressed genes are regulated directly by PRDM16 in vivo. By manual inspection, they find that "most" of these show a PRDM16 peak. How many is most? If using the same parameters for determining peaks, how many genes in an appropriately chosen negative control set of genes would show peaks? Can the authors rigorously establish the statistical significance of this observation? And why wasn't the same experiment performed on the NSCs in which the other experiments are done so one can directly compare the results? Instead, as far as I could tell, there is only ChIP-qPCR for two genes in NSCs in Supplementary Figure 4D.

In our text, we indicated the genes containing PRDM16 binding peaks in the figures and described them as “Text in black in Fig. 6A and Supplementary Fig. 5A”. We will add the precise number “25 of these genes” in the main text to clarify it. We used BMP-only repressed 184-31 = 153 genes (excluding PRDM16-BMP4 co-repressed) as a negative control set of genes. By computationally determine the nearest TSS to a PRDM16 peak, we identified 24/31 co-repressed genes and 84/153 BMP-only-repressed genes, containing PRDM16 peaks in the E12.5 ChP data. Fisher’s Exact Test comparing the proportions yields the P-value = 0.015.

We are confused with the second part of the comment “And why wasn't the same experiment performed on the NSCs in which the other experiments are done so one can directly compare the results? Instead, as far as I could tell, there is only ChIP-qPCR for two genes in NSCs in Supplementary Figure 4D.” If the reviewer meant why we didn’t sequence the material from sequential-ChIP or validate more taget genes, the reason is the limitation of the material. Sequential ChIP requires a large quantity of the antibodies, and yields little material barely sufficient for a few qPCR after the second round of IP. This yielded amount was far below the minimum required for library construction. The PRDM16 antibody was a gift, and the quantity we have was very limited. We made a lot of efforts to optimize all available commercial antibodies in ChIP and Cut&Tag, but none of them worked in these assays.

(6) In comparing RNA in situ between WT and PRDM16 KO in Figure 7, the authors state they use the Wnt2b signal to identify the border between CH and neocortex. However, the Wnt2b signal is shown in grey and it is impossible for this reviewer to see clear Wnt2b expression or where the boundaries are in Figure 7A. The authors also do not show where they placed the boundaries in their analysis. Furthermore, Figure 7B only shows insets for one of the regions being compared making it difficult to see differences from the other region. Finally, the authors do not show an example of their spot segmentation to judge whether their spot counting is reliable. Overall, this makes it difficult to judge whether the quantification in Figure 7C can be trusted.

In the revised manuscript we have included an individal channel of Wnt2b and mark the boundaries. We also provide full-view images and examples of spot segmentation in the new supplementary figure 8.

(7) The correlation between mKi67 and Axin2 in Figure 7 is interesting but does not convincingly show that Wnt downstream of PRDM16 and BMP is responsible for the increased proliferation in PRDM16 mutants.

We agree that this result (the correlation between mKi67 and Axin2) alone only suggests that Wnt signaling is related to the proliferation defect in the Prdm16 mutant, and does not necessarily mean that Wnt is downstream of PRDM16 and BMP. Our concolusion is backed up by two additional lines of evidences: the Cut&Tag data in which PRDM16 binds to regulatory regions of Wnt7b and Wnt3a; BMP and PRDM16 co-repress Wnt7b in vitro.

An ideal result is that down-regulating Wnt signaling in Prdm16 mutant can rescue Prdm16 mutant phenotype. Such an experiment is technically challenging. Wnt plays diverse and essential roles in NSC regulation, and one would need to use a celltype-and stage-specific tool to down-regulate Wnt in the background of Prdm16 mutation. Moreover, Wnt genes are not the only targets regulated by PRDM16 in these cells, and downregulating Wnt may not be sufficient to rescue the phenotype.

Weaknesses of the presentation:Overall, the manuscript is not easy to read. This can cause confusion.

We have revised the text to improve clarity.

**Reviewer #1 (Recommendations for the authors):**
(1) Overall, the manuscript is not easy to read. Here are some causes of confusion for which the presentation could be cleaned up:

We are grateful for the reviewer’s suggestion. In the revised manuscript, we have made efforts to improve the clarity of the text.

(a) Part of the first section is confusing in that some statements seem contradictory, in particular:"there is no overall patterning defect of ChP and CH in the Prdm16 mutant" (line 125)"Prdm16 depletion disrupted the transition from neural progenitors into ChP epithelia" (line 144)It would be helpful if the authors could reformulate this more clearly.

We modified the text to clarify that while the BMP-patterned domain is not affected, the transition of NSCs into ChP epithelial cells is compromised in the Prdm16 mutant.

(b) Flag_PRDM16, PRDM16_expressing, PRDM16_E, PRDM16 OE all seem to refer to the same PRDM16 overexpressing cells, which is very confusing. The authors should use consistent naming. Moreover, it would be good if they renamed these all to PRDM16_OE to indicate expression is not endogenous but driven by a constitutive promoter.

We appreciate the comment and agree that the use of multiple terms to refer to the same PRDM16-overexpressing condition was confusing. Our original intention in using Prdm16_E was to distinguish cells expressing PRDM16 from the two other groups: wild-type cells and Prdm16_KO cells, which both lack PRDM16 protein expression. However, we acknowledge that Prdm16_E could be misinterpreted as indicating expression from the endogenous Prdm16 promoter. To avoid this confusion and ensure consistency, we have now standardized the terminology and refer to this condition as Prdm16_OE, indicating Flag-tagged PRDM16 expression driven by a constitutive promoter.

(c) Line 179 states "generated a cell line by infecting Prdm16_KO cells with the same viral vector, expressing 3xNSL_Flag". Do the authors mean 3xNLS_Flag_Prdm16, so these are the Prdm16_KO_E cells by the notation suggested above? Or is this a control vector with Flag only? The following paragraph refers to Supplementary Figure 2C-F where the same construct is called KO_CDH, suggesting this was an empty CDH vector, without Flag, or Prdm16. This is confusing.

We appreciate the reviewer’s careful reading and helpful comment. We acknowledge the confusion caused by the inconsistent terminology. To clarify: in line 179, we intended to describe an attempt to generate a Prdm16_KO cell line expressing 3xNLS_Flag_Prdm16, not a control vector with Flag only. However, despite repeated attempts, we were unable to establish this line due to low viral efficiency and the vulnerability of Prdm16_KO cells to infection with the large construct. Therefore, these cells were not included in the subsequent analyses.

The term KO_CDH refers to Prdm16_KO cells infected with the empty CDH control vector, which lacks both Flag and Prdm16. This is the line used in the experiments shown in Supplementary Fig. 2C–F. We have revised the text throughout the manuscript to ensure consistent use of terminology and to avoid this confusion.

(2) The introductory statements on lines 53-54 could use more references.

Thanks for the suggestion. We have now included more references.

(3) It would be helpful if all structures described in the introduction and first section were annotated in Figure 1, or otherwise, if a cartoon were included. For example, the cortical hem, and fourth ventricle.

Thanks for the suggestion. We have now indicated the structures, ChP, CH and the fourth ventricle, in the images in Figure 1 and Supplementary Figure 1.

(4) In line 115, "as previously shown" - to keep the paper self-contained a figure illustrating the genetics of the KO allele would be helpful.

Thanks for the suggestion. We have now included an illustration of the Prdm16 cGT allele in Figure 1B.

(5) In Figure 1D as costain for a ChP marker would be helpful because it is hard to identify morphologically in the Prdm16 KO.

Appoligize for the unclarity. The KO allele contains a b-geo reporter driven by Prdm16 endogenous promoter. The samples were co-stained for EdU, b-Gal and DAPI. To distingquish the ChP domain from the CH, we used the presence of b b-Gal as a marker. We indicated this in the figure legend, but now have also clarified this in the revised text.

(6) The details in Figure 1E are hard to see, a zoomed-in inset would help.

A zoomed-in inset is now included in the figure.

(7) Supplementary Figure 2A does not convincingly show that PRDM16 protein is undetectable since endogenous expression may be very low compared to the overexpression PRDM16_E cells so if the contrast is scaled together it could appear black like the KO.

We appreciate the reviewer’s point and have carefully considered this concern. We concluded that PRDM16 protein is effectively undetectable in cultured wild-type NSCs based on direct comparison with brain tissue. Both cultured NSCs and brain sections were processed under similar immunostaining and imaging conditions. While PRDM16 showed robust and specific nuclear localization in embryonic brain sections (Fig. 1B and Supplementary Fig. 1A), only a small subset of cultured NSCs exhibited PRDM16 signal, primarily in the cytoplasm (middle panel of Fig. 2A). This stark contrast supports our conclusion that endogenous PRDM16 protein is either absent or significantly downregulated in vitro. Because of this limitation, we turned to over-expressing Prdm16 in NSC culture using a constitutive promoter.

(9) Line 182 "Following the washout step" - no such step had been described, maybe replace by "After washout of BMP".

Yes, we have revised the text.

(8) Line 214: "indicating a modest level" - what defines modest? Compared to what? Why is a few thousand moderate rather than low? Does it go to zero with inhibitors for pathways?

Here a modest level means a lower level than to that after adding BMP4. To clarify this, we revised the description to “indicating endogenous levels of …”

(9) The way qPCR data are displayed makes it difficult to appreciate the magnitude of changes, e.g. in Supplementary Figure 2B where a gap is introduced on the scale. Displaying log fold change / relative CT values would be more informative.

We used a segmented Y-axis in Supplementary Figure 2B because the Prdm16 overexpression samples exhibited much higher experssion levels compared to other conditions. In response to this suggestion, we explored alternative ways to present the result, including ploting log-transformed values and log fold changes. However, these methods did not enhance the clarity of the differences – in fact, log scaling made the magnitude of change appear less apparent. To address this, we now present the overexpression samples in a separate graph, thereby eliminating the need for a broken Y-axis and improving the overall readability of the data.

(10) Writing out "3 days" instead of 3D in Figure 2A would improve clarity. It would be good if the used time interval is repeated in other figures throughout the paper so it is still clear the comparison is between 0 and 3 days.

We have changed “3D” to “3 days”. All BMP4 treatments in this study were 3 days.

(11) Line 290: "we found that over 50% of SMAD4 and pSMAD1/5/8 binding peaks were consistent in Prdm16_E and Prdm16_KO cells, indicating that deletion of Prdm16 does not affect the general genomic binding ability of these proteins" - this only makes sense to state with appropriate controls because 50% seems like a big difference, what is the sample to sample variability for the same condition? Moreover, the next paragraph seems to contradict this, ending with "This result suggests that SMAD binding to these sites depends on PRDM16". The authors should probably clarify the writing.

We appreciate the reviwer’s comment and agree that clarification was needed. Our point was that SMAD4 and pSMAD1/5/8 retain the ability to bind DNA broadly in the Prdm16 KO cells, with more than half of the original binding sites still occupied. This suggests that deletion of Prdm16 does not globally impair SMAD genomic binding. Howerever, our primary interest lies in the subset of sites that show differential by SMAD binding between wt and Prdm16 KO conditions, as thse are likely to be PRDM16-dependent.

In the following paragraph, we focused specifically on describing SMAD and PRDM16 co-bound sites. At these loci, SMAD4 and pSMAD1/5/8 showed reduced enrichment in the absence of PRDM16, suggesting PRDM16 facilitates SMAD binding at these particular regions. We have revised the text in the manuscript to more clearly distinguish between global SMAD binding and PRDM16-dependent sites.

(12) Much more convincing than ChIP-qPCR for c-FOS for two loci in Figures 5F-G would be a global analysis of c-FOS ChIP-seq data.

We agree that a global c-FOS ChIP-seq analysis would provide a more comprehensive view of c-FOS binding patterns. However, the primary focus of this study is the interaction between BMP signaling and PRDM16. The enrichment of AP-1 motifs at ectopic SMAD4 binding sites was an unexpected finding, which we validated using c-FOS ChIP-qPCR at selected loci. While a genome-wide analysis would be valuable, it falls beyond the current scope. We agree that future studies exploring the interplay among SMAD4/pSMAD, PRDM16, and AP-1 will be important and informative.

(13) Figure 6A is hard to read. A heatmap would make it much easier to see differences in expression. Furthermore, if the point is to see the difference between ChP and CH, why not combine the different subclusters belonging to those structures? Finally, why are there 28 genes total when it is said the authors are evaluating a list of 31 genes and also displaying 6 genes that are not expressed (so the difference isn't that unexpressed genes are omitted)?

For the scRNA-seq data, we chose violin plots because they display both gene expression levels and the number of cells that express each gene. However, we agree that the labels in Figure 6A were too small and difficult to read. We have revised the figure by increasing the font size and moved genes with low expression to Supplementary Figure 5A. Figure 6A includes 17 more highly expressed genes together with three markers, and Supplementary Figure 5A contains 13 lowly expressed genes. One gene Mrtfb is missing in the scRNA-seq data and thus not included. We have revised the description of the result in the main text and figure legends.

**Reviewer #2 (Public review):**
Summary:This article investigates the role of PRDM16 in regulating cell proliferation and differentiation during choroid plexus (ChP) development in mice. The study finds that PRDM16 acts as a corepressor in the BMP signaling pathway, which is crucial for ChP formation.The key findings of the study are:(1) PRDM16 promotes cell cycle exit in neural epithelial cells at the ChP primordium.(2) PRDM16 and BMP signaling work together to induce neural stem cell (NSC) quiescence in vitro.(3) BMP signaling and PRDM16 cooperatively repress proliferation genes.(4) PRDM16 assists genomic binding of SMAD4 and pSMAD1/5/8.(5) Genes co-regulated by SMADs and PRDM16 in NSCs are repressed in the developing ChP.(6) PRDM16 represses Wnt7b and Wnt activity in the developing ChP.(7) Levels of Wnt activity correlate with cell proliferation in the developing ChP and CH.In summary, this study identifies PRDM16 as a key regulator of the balance between BMP and Wnt signaling during ChP development. PRDM16 facilitates the repressive function of BMP signaling on cell proliferation while simultaneously suppressing Wnt signaling. This interplay between signaling pathways and PRDM16 is essential for the proper specification and differentiation of ChP epithelial cells. This study provides new insights into the molecular mechanisms governing ChP development and may have implications for understanding the pathogenesis of ChP tumors and other related diseases.Strengths:(1) Combining in vitro and in vivo experiments to provide a comprehensive understanding of PRDM16 function in ChP development.(2) Uses of a variety of techniques, including immunostaining, RNA in situ hybridization, RT-qPCR, CUT&Tag, ChIP-seq, and SCRINSHOT.(3) Identifying a novel role for PRDM16 in regulating the balance between BMP and Wnt signaling.(4) Providing a mechanistic explanation for how PRDM16 enhances the repressive function of BMP signaling. The identification of SMAD palindromic motifs as preferred binding sites for the SMAD/PRDM16 complex suggests a specific mechanism for PRDM16-mediated gene repression.(5) Highlighting the potential clinical relevance of PRDM16 in the context of ChP tumors and other related diseases. By demonstrating the crucial role of PRDM16 in controlling ChP development, the study suggests that dysregulation of PRDM16 may contribute to the pathogenesis of these conditions.

We thank the reviewer for the thorough and thoughtful summary of our study. We’re glad the key findings and significance of our work were clearly conveyed, particularly regarding the role of PRDM16 in coordinating BMP and Wnt signaling during ChP development. We also appreciate the recognition of our integrated approach and the potential implications for understanding ChP-related diseases.

Weaknesses:(1) Limited investigation of the mechanism controlling PRDM16 protein stability and nuclear localization in vivo. The study observed that PRDM16 protein became nearly undetectable in NSCs cultured in vitro, despite high mRNA levels. While the authors speculate that post-translational modifications might regulate PRDM16 in NSCs similar to brown adipocytes, further investigation is needed to confirm this and understand the precise mechanism controlling PRDM16 protein levels in vivo.

While mechansims controlling PRDM16 protein stability and nuclear localization in the developing brain are interesting, the scope of this paper is revealing the function of PRDM16 in the choroid plexus and its interaction with BMP signaling. We will be happy to pursuit this direction in our next study.

(2) Reliance on overexpression of PRDM16 in NSC cultures. To study PRDM16 function in vitro, the authors used a lentiviral construct to constitutively express PRDM16 in NSCs. While this approach allowed them to overcome the issue of low PRDM16 protein levels in vitro, it is important to consider that overexpressing PRDM16 may not fully recapitulate its physiological role in regulating gene expression and cell behavior.

As stated above, we acknowledge that findings from cultured NSCs may not directly apply to ChP cells in vivo. We are cautious with our statements. The cell culture work was aimed to identify potential mechanisms by which PRDM16 and SMADs interact to regulate gene expression and target genes co-regulated by these factors. We expect that not all targets from cell culture are regulated by PRDM16 and SMADs in the ChP, so we validated expression changes of several target genes in the developing ChP and now included the new data in Fig. 7 and Supplementary Fig. 7. Out of the 31 genes identified from cultured cells, four cell cycle regulators including Wnt7b, Id3, Spc24/25/nuf2 and Mybl2, showed de-repression in Prdm16 mutant ChP. These genes can be relevant downstream genes in the ChP, and other target genes may be cortical NSC-specific or less dependent on Prdm16 in vivo.

(3) Lack of direct evidence for AP1 as the co-factor responsible for SMAD relocation in the absence of PRDM16. While the study identified the AP1 motif as enriched in SMAD binding sites in Prdm16 knockout cells, they only provided ChIP-qPCR validation for c-FOS binding at two specific loci (Wnt7b and Id3). Further investigation is needed to confirm the direct interaction between AP1 and SMAD proteins in the absence of PRDM16 and to rule out other potential co-factors.

We agree that the finding of the AP1 motif enriched at the PRDM16 and SMAD co-binding regions in Prdm16 KO cells can only indirectly suggest AP1 as a co-factor for SMAD relocation. That’s why we used ChIP-qPCR to examine the presence of C-fos at these sites. Although we only validated two targets, the result confirms that C-fos binds to the sites only in the Prdm16 KO cells but not Prdm16_expressing cells, suggesting AP1 is a co-factor. Our results cannot rule out the presence of other co-factors.

**Reviewer #2 (Recommendations for the authors):**
Minor typo: [7, page 3] "sicne" should be "since".

We appreciate the reviewer’s careful reading. We have now corrected the typo and revised some part of the text to improve clarity.

**Reviewer #3 (Public review):**
Summary:Bone morphogenetic protein (BMP) signaling instructs multiple processes during development including cell proliferation and differentiation. The authors set out to understand the role of PRDM16 in these various functions of BMP signaling. They find that PRDM16 and BMP co-operate to repress stem cell proliferation by regulating the genomic distribution of BMP pathway transcription factors. They additionally show that PRDM16 impacts choroid plexus epithelial cell specification. The authors provide evidence for a regulatory circuit (constituting of BMP, PRDM16, and Wnt) that influences stem cell proliferation/differentiation.Strengths:I find the topics studied by the authors in this study of general interest to the field, the experiments well-controlled and the analysis in the paper sound.

We thank the reviewer for their positive feedback and thoughtful summary. We appreciate the recognition of our efforts to define the role of PRDM16 in BMP signaling and stem cell regulation, as well as the soundness of our experimental design and analysis.

Weaknesses:I have no major scientific concerns. I have some minor recommendations that will help improve the paper (regarding the discussion).

We have revised the discussion according to the suggestions.

**Reviewer #3 (Recommendations for the authors):**
Specific minor recommendations:Page 18. Line 526: In a footnote, the authors point out a recent report which in parallel was investigating the link between PRDM16 and SMAD4. There is substantial non-overlap between these two papers. To aid the reader, I would encourage the authors to discuss that paper in the discussion section of the manuscript itself, highlighting any similarities/differences in the topic/results.

Thanks for the suggestion. We now included the comparison in the discussion. One conclusion between our study and this publication is consistent, that PRDM16 functions as a co-repressor of SMAD4. However, the mechanims are different. Our data suggests a model in which PRDM16 facilitates SMAD4/pSMAD binding to repress proliferation genes under high BMP conditions. However, the other report suggests that SMAD4 steadily binds to Prdm16 promoter and switches regulatory functions depending on the co-factors. Together with PRDM16, SMAD4 represses gene expression, while with SMAD3 in response to high levels of TGF-b1, it activates gene expression. These differences could be due to different signaling (BMP versus TGF-b), contexts (NSCs versus Pancreatic cancers) etc.

Page 3. Line 65: typo 'since'

We appreciate the reviewer’s careful reading. We have now corrected the typo and revised the text to improve clarity.